# Measuring vaccination coverage and concerns of vaccine holdouts from web search logs

Serina Chang[1], Adam Fourney[2] & Eric Horvitz[2] ✉

To design effective vaccine policies, policymakers need detailed data about who has been vaccinated, who is holding out, and why. However, existing data in the US are insufficient: reported vaccination rates are often delayed or not granular enough, and surveys of vaccine hesitancy are limited by high-level questions and self-report biases. Here we show how search engine logs and machine learning can help to fill these gaps, using anonymized Bing data from February to August 2021. First, we develop a *vaccine intent classifier* that accurately detects when a user is seeking the COVID-19 vaccine on Bing. Our classifier demonstrates strong agreement with CDC vaccination rates, while preceding CDC reporting by 1–2 weeks, and estimates more granular ZIP-level rates, revealing local heterogeneity in vaccine seeking. To study vaccine hesitancy, we use our classifier to identify two groups, *vaccine early adopters* and *vaccine holdouts*. We find that holdouts, compared to early adopters matched on covariates, are 67% likelier to click on untrusted news sites, and are much more concerned about vaccine requirements, development, and vaccine myths. Even within holdouts, clusters emerge with different concerns and openness to the vaccine. Finally, we explore the temporal dynamics of vaccine concerns and vaccine seeking, and find that key indicators predict when individuals convert from holding out to seeking the vaccine.

COVID-19 vaccines provide significant protection against severe cases of SARS-CoV-2[1,2], yet a large portion of the United States remains unvaccinated. Effective vaccine policies–for example, where to place vaccine sites[3,4], how to communicate about the vaccine[5,6], and how to design campaigns to reach unvaccinated populations[7–9]–rely on detailed data about who is seeking vaccination, who is holding out, and why. However, existing data are insufficient[10]. Reported vaccination rates are frequently delayed[11], missing at the county-level and below[12], and missing essential demographic data[13,14]. Surveys provide a starting point for understanding vaccine hesitancy but are often limited by high-level questions[15], small or biased samples[16,17], and self-reporting biases (e.g., recall or social desirability bias)[18,19], especially in sensitive contexts such as vaccination[20].

Here we show how large-scale search engine logs and machine learning can be leveraged to fill these gaps, enabling fine-grained estimation of vaccine rates and discovering the concerns of vaccine holdouts from their search interests. We use billions of anonymized search logs from Bing and introduce two computational resources to extract meaning from unlabeled queries and clicks. First, we develop a *vaccine intent classifier* to detect when a user is seeking the COVID-19 vaccine on search. Our classifier achieves areas under the receiver operating characteristic curve (AUCs) above 0.90 in all 50 states, and demonstrates strong agreement with CDC vaccination rates across states ($r = 0.86$) and over time ($r = 0.89$). Using our classifier, we can estimate vaccine intent rates to the level of ZIP code tabulation areas (ZCTAs), producing the most comprehensive dataset to-date of ZIP-

[1]Department of Computer Science, Stanford University, Stanford, CA, USA. [2]Microsoft Research, Redmond, WA, USA. ✉e-mail: horvitz@microsoft.com

level estimates of COVID-19 vaccination or vaccine intent. Our search signals precede CDC reporting by 1–2 weeks and provide more granular information, as ZCTAs are 10x the granularity of the CDC's county-level data. Our second resource is a novel *taxonomy of COVID-19 vaccine concerns* on search. Our taxonomy consists of 25,000 vaccine-related URLs, clicked on by Bing users, that we organized into a hierarchy of vaccine concerns from eight top categories to 36 sub-categories to 156 low-level topics. Unlike surveys, our taxonomy discovers these concerns directly from users' expressed interests and explores them at multiple scales. Furthermore, by measuring individuals' interest in each concern from their clicks, we capture revealed preferences, side-stepping potential biases in self-reporting[19,21]. We have publicly released our vaccine intent estimates, from ZCTA to state-level, and taxonomy of vaccine concerns, along with our code.

Combining our taxonomy with the vaccine intent classifier allows us to conduct a thorough analysis of how individuals' vaccine concerns relate to whether they decide to seek the vaccine. We use our classifier to identify two groups of users—*vaccine early adopters* and *vaccine holdouts*—and compare their search behaviors. We find that vaccine holdouts, compared to early adopters matched on covariates, are 67% (95% CI, 66%–68%) more likely to click on untrusted news sites, and are far more concerned with vaccine requirements, vaccine development and approval, and vaccine myths. We also find that vaccine concerns differ significantly within holdouts, across demographic groups and through clustering of individual concerns, where we discover four distinct holdout profiles who differ in their key concerns and openness to the COVID-19 vaccine. Finally, we analyze the temporal dynamics of vaccine concerns and vaccine seeking, and find that individuals exhibit telltale shifts in their vaccine concerns when they eventually convert from holding out to seeking the vaccine. Our findings demonstrate the need for policymakers to go beyond one-size-fits-all solutions, so that messaging is tailored to each individual's unique vaccine concerns and how close they are to seeking the vaccine.

We use Bing search logs, which have been used to study other health issues such as shifts in needs and disparities in information access during the pandemic[22,23], health information needs in developing nations[24], experiences around cancer diagnoses[25,26], concerns during pregnancy[27], nutritional patterns across the world[28], and medical anxieties associated with online search[29]. Our efforts build on prior work that extracts insights about the COVID-19 vaccine from digital traces, such as social media[30–32] and aggregated search trends[33–35], and other efforts to detect health conditions online, such as depression[36] or flu[37]. Our work seeks to address the challenges of working with digital traces[21,38] and limitations of prior work[37,39] by developing rigorous, human-in-the-loop methods to precisely detect user intents and search interests ("Vaccine intent classifier" and "Taxonomy of vaccine concerns on search"), correct for bias from non-uniform Bing coverage ("Coverage-corrected vaccine intent rates"), and validate our results against external data ("Comparison to reported vaccination rates"). By leveraging search logs and machine learning, our approach provides real-time, fine-grained signals about vaccine seeking and holding out, and nuanced understandings of vaccine concerns, helping to guide more timely and effective vaccine policies.

## Methods overview: vaccine intent classifier

We develop a machine learning classifier to detect when users are expressing vaccine intent, i.e., seeking the COVID-19 vaccine on search. Vaccine intent can be expressed through unambiguous queries, such as [covid vaccine near me], which we detect using regular expressions that specify patterns to match in text. However, vaccine intent can also be clarified through clicks on search results[40]: for example, a user may issue an ambiguous query, such as [covid vaccine], then clarify their intent by clicking on the URL for the CVS COVID-19 vaccine registration page. The challenge with URLs is that they are less formulaic than queries, so we cannot easily define regular expressions to identify

URLs expressing vaccine intent. Instead, we employ a series of graph-based machine learning techniques, combined with manual annotation, to identify URLs.

Our key insight is that, while we cannot use regular expressions to identify URLs, we can use them to identify vaccine intent queries and then use those queries to identify URLs, based on common query-click patterns. For example, vaccine intent queries such as [cvs covid vaccine] or [covid vaccine near me] may result in clicks on the CVS COVID-19 vaccine registration page. To capture these patterns, we construct large-scale *query-click graphs*[41,42], which are bipartite networks between queries and URLs where an edge from a query to a URL indicates how often this query is followed by a click on this URL (Fig. 1a). We use regular expressions to identify vaccine intent queries in the graph, and then propagate labels from these queries to URLs via Personalized PageRank (Fig. 1b, left)[43,44]. This process enables us to identify URL candidates that likely express vaccine intent, without any URL labels.

We then present the URL candidates to annotators on Amazon Mechanical Turk and ask them to label whether these URLs indicate vaccine intent (Fig. 1b, middle). We observe strong performance from our PageRank-based approach: even if positive labels require agreement from 3 annotators (out of 3-4), we find that 86% of the URL candidates are labeled positive for vaccine intent (Fig. S2). However, since manual annotation is expensive, we are only able to label around 2000 URLs through this method. To expand this set, we use these labels to train graph neural networks[45] (GNNs) to predict vaccine intent, so that we can use GNNs to predict labels for the remaining URLs (Fig. 1b, right). Our GNNs demonstrate strong performance in all 50 states, with AUCs over 0.90 on held-out URLs labeled for vaccine intent (Fig. S4). Using our GNNs, we discover 11,400 more URLs that are highly indicative of vaccine intent.

## Correcting for bias in vaccine intent estimates

We apply our classifier to Bing search logs from Feburary 1 to August 31, 2021 ("Datasets") and identify 7.45 million active Bing users who have expressed vaccine intent through their queries or clicks. However, before we can use the classifier to estimate regional rates of vaccine intent, we need to correct for potential sources of bias in our approach. We decompose potential bias into two key sources ("Decomposition of bias"): first, bias from non-uniform Bing coverage, and second, bias from non-uniform true and false positive rates of our classifier. By correcting for non-uniform Bing coverage ("Coverage-corrected vaccine intent rates") and demonstrating that our classifier's true and false positive rates do not significantly differ across regions ("Bias in vaccine intent classifier"), our vaccine intent estimates should, theoretically, form unbiased estimates of true vaccination rates. Supporting this claim are our empirical results showing that our vaccine intent estimates agree strongly with CDC vaccination rates. Furthermore, to evaluate the representativeness of Bing data, we compare search trends for vaccine intent queries between Google and Bing and find that, even before applying corrections to Bing data, the trends are highly correlated (Figs. S9–S10).

## Results
### Our vaccine intent estimates are highly correlated with CDC data

When we compare our vaccine intent estimates to state-level vaccination rates from the CDC, we observe strong correlation ($r = 0.86$) on cumulative rates at the end of August 2021 (Fig. 1c). Notably, we find that the correlation drops to $r = 0.79$ if we do not correct for Bing coverage in our estimates. If we only use queries to detect vaccine intent, the correlation drops to $r = 0.62$ and we lose 57% of the users we identified with our full classifier, demonstrating the value of including URLs (Table 1). Additionally, we compare our vaccine intent estimates to the CDC's vaccination rates over time. We observe strong correlations here as well, especially if we allow the CDC time series to lag

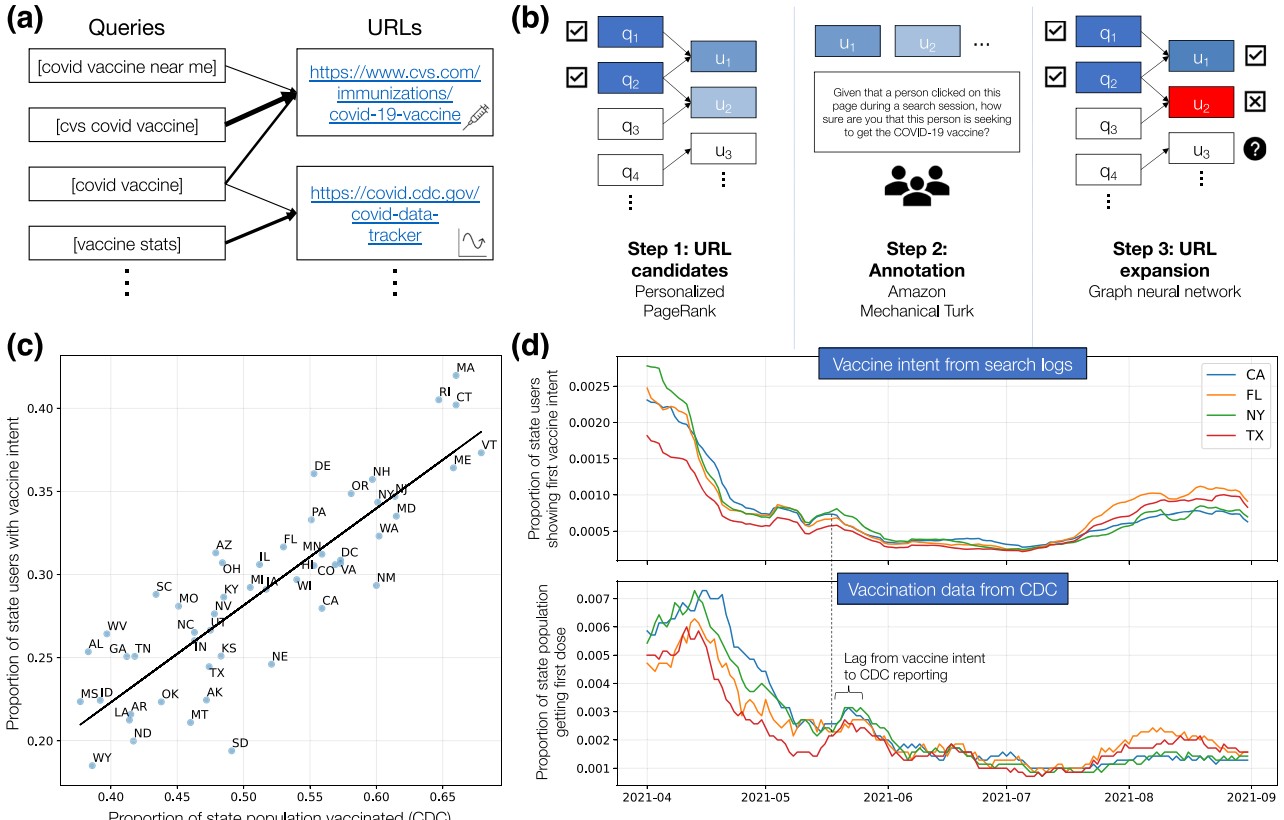

**Fig. 1 | Vaccine intent classifier. a** Our computational approach centers on query-click graphs constructed from billions of Bing search logs. **b** Using these graphs, we introduce a three-step pipeline to identify vaccine intent URLs: generate URL candidates via Personalized PageRank; present URL candidates to annotators; and expand the final set of URLs with graph neural networks. Each step improves our coverage of users and correlation with CDC vaccination rates (Table 1). **c** Our vaccine intent estimates are highly correlated with state vaccination rates from the CDC. Here, we compare cumulative rates up to August 31, 2021 ($r = 0.86$). **d** Our estimates are also highly correlated with CDC rates over time ($r = 0.89$, median over states), with the CDC time series lagging by 7 and 15 days (IQR). Here, we visualize time series for the 4 largest states in the US, with extended results in "Comparison to reported vaccination rates".

behind the vaccine intent time series. With lags of 7–15 days (IQR), the median correlation over states reaches $r = 0.89$; without a lag, the median correlation drops to $r = 0.78$. The CDC's lag demonstrates an advantage of the classifier, as it can detect vaccine seeking in real time without delays from reporting. Furthermore, our vaccine intent rates help to forecast daily vaccinations, significantly improving predictive performance over only using past values of daily vaccinations (i.e., with a Granger-causal interpretation) in the majority of US states (Table S4). Thus, our vaccine intent rates both precede and predict vaccinations, showing promise as a signal that could help policymakers better pre-empt vaccine demand and meet population needs.

### Granular trends in vaccine seeking

Our vaccine intent classifier allows us to pinpoint who was seeking the COVID-19 vaccine, where, and when. We estimate cumulative vaccine intent rates up to the end of August 2021 at the level of ZCTAs (Fig. 2a),

**Table 1 | Each step of our classification pipeline ("Vaccine intent classifier") improves both our correlation with CDC state vaccination rates and our coverage of vaccine intent users**

| Pipeline step | Correlation with CDC | Num vaccine intent users |
| --- | --- | --- |
| Only queries | 0.62 | 3.18M |
| +manual URLs | 0.80 | 4.95M |
| +manual and GNN URLs | 0.86 | 7.45M |

approximately 10x the granularity of counties, which is the finest-grained vaccination data the CDC provides and, still, with many counties missing or having incomplete data[12]. We observe substantial heterogeneity in vaccine intent at the ZCTA-level, even within the same states and counties. For example, when we focus on New York City, we see that Manhattan and Queens have higher vaccine intent rates, and within Queens, ZCTAs in the northern half have higher rates (Fig. 2b), aligning with reported local vaccination rates in New York City[46]. In fact, we show that variation in vaccine intent rates within counties often exceeds variation between counties (Fig. S11), motivating the need for finer-grained estimates of vaccine rates. We can also use our estimates to characterize demographic trends in vaccination. When we measure correlations between ZCTA vaccine intent rate and different demographic variables, we find that overall demographic trends from our estimates align closely with prior literature[16,47–49]. For example, we observe strong positive correlations with education, income, and population density, and a strong negative correlation with percent Republican (Fig. 2c). However, we discover more nuanced trends when we look closer: demographic trends vary significantly across states (Fig. S12), especially for race and ethnicity, and trends change over time (Fig. S18). Thus, our classifier both confirms existing findings and enables new analyses with finer granularity across regions, demographics, and time.

### Taxonomy of vaccine concerns on search

To characterize vaccine-related search interests, we construct a hierarchical taxonomy of vaccine concerns, defined in terms of 25,000

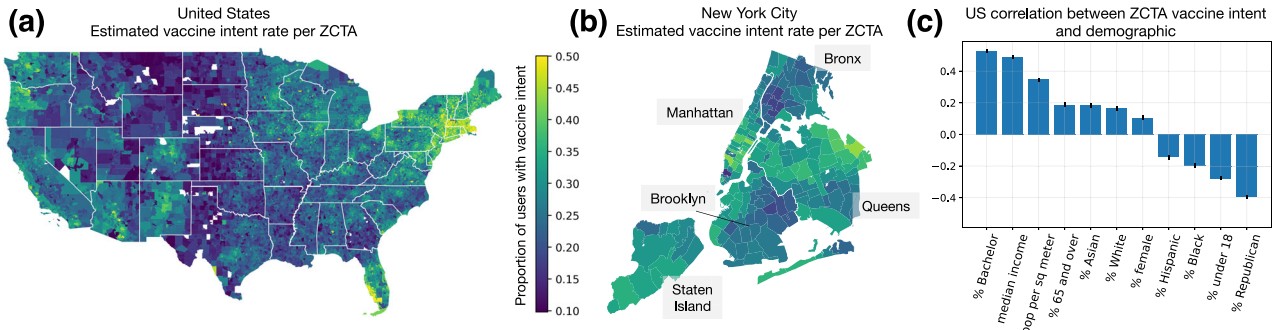

**Fig. 2 | Granular trends in vaccine seeking. a** Using our classifier, we can estimate vaccine intent rates per ZIP code tabulation area (ZCTA), approximately 10x the granularity of counties. **b** Zooming in on New York City shows that estimated vaccine intent rates vary substantially across ZCTAs, even within the same city or county. **c** To characterize demographic trends in vaccination, we measure Pearson correlations between ZCTA vaccine intent rates and demographic variables, over $N = 20,899$ ZCTAs. Error bars indicate 95% CIs.

vaccine-related URLs (i.e., containing "vaccin" or "vax"). First, using the Louvain algorithm for community detection on graphs[50], we automatically partition the URLs into 156 clusters, each containing around 100–500 URLs. Based on these clusters, which are remarkably coherent (Table S5), we design a comprehensive set of subcategories and top categories, and assign the clusters accordingly. For example, we identify one cluster of news stories announcing vaccine passport requirements in cities such as New York City and Los Angeles[51,52], which we assign to the proof of vaccination subcategory and Vaccine Requirements top category. This bottom-up approach allows us to discover and measure vaccine concerns directly from users' search interests and analyze them at multiple scales, providing complementary insights to more traditional surveys. In Fig. 3, we summarize our resulting taxonomy, which consists of 8 top categories and 36 subcategories. Some top categories encompass a number of distinct subcategories: for example, under Vaccine Safety, we include normal side effects, severe side effects, concerns about reproductive health, vaccine history and development, FDA approval, fear of vaccine-caused deaths, and eerie fears (e.g., myths about vaccine shedding or becoming magnetic[53]).

### Vaccine holdouts and early adopters

We use our vaccine intent classifier to identify two groups: *vaccine early adopters*, who expressed their first vaccine intent before May 2021, and *vaccine holdouts*, who waited until July 2021 to show their first vaccine intent, despite becoming eligible by April[54]. Comparing the search behavior of these two groups allows us to discover relationships between vaccine seeking, vaccine concerns, and news consumption. To reduce potential confounding, we match each holdout with a unique early adopter from the same county and with a similar average query count, since we know that the populations seeking vaccination changed over time and we do not want our comparisons to be overpowered by regional or demographic differences. In our following analyses, we compare the search interests of the matched sets, with over 200,000 users in each set.

First, we analyze the trustworthiness of news sites clicked on by vaccine holdouts versus early adopters. We use ratings from Newsguard, which assigns trust scores to news sites based on criteria such as how often the site publishes false content and how it handles the difference between news and opinion[55]. We find that, in the period while vaccine holdouts were eligible but still holding out (April to June 2021), holdouts were 67% (95% CI, 66%–68%) likelier than their matched early adopters to click on untrusted news, defined by Newsguard as domains with trust scores below 60. Furthermore, we see that as the trust score from Newsguard degrades, the likelier it was that holdouts clicked on the site, relative to early adopters (Fig. 4a), with a negative correlation of $r = −0.41$. For example, sites that are known for

spreading COVID-19 misinformation, such as infowars.com[56], RT.com[57], and mercola.com[58], were much likelier to be clicked on by holdouts. On the other hand, the negative relationship between trust score and relative likelihood of holdouts clicking only becomes stronger for mainstream news ($r = −0.56$), showing that this relationship is not driven by fringe, outlier news domains. These results extend prior work linking vaccine hesitancy and misinformation[30,31,59] by measuring news consumption more directly (through search clicks instead of posts on social media) and showing that holdouts are still consuming vastly different news from early adopters, even after controlling for regional politics (by matching on county).

### Distinctive vaccine concerns of holdouts

Using our taxonomy of vaccine concerns, we find that at the top category-level, vaccine holdouts are the most concerned about Vaccine Safety, which accounts for 23% of their vaccine-related clicks, followed by Vaccine Information (10%) and Vaccine Requirements (9%). We also observe changes in interests over time (Fig. 4b): for example, interest in Vaccine Incentives increased in May 2021, when incentives were introduced[60], and interest in Vaccine Effectiveness grew in June, following the spread of the Delta variant.

We also compare the vaccine concerns of holdouts and their matched early adopters. From April to June 2021, we find that holdouts were 48% less likely than early adopters to click on any vaccine-related URL. Furthermore, their distribution of concerns within their vaccine-related clicks differed significantly (Fig. 4c, Table S6). Using the subcategories from the taxonomy, we find that holdouts were far more interested in religious concerns about the vaccine; anti-vaccine messages from experts and high-profile figures; avoiding vaccine requirements by seeking exemptions, banning mandates, or obtaining fake proof of vaccination; eerie fears and vaccine-caused deaths; and FDA approval and vaccine development. In comparison, early adopters were much more concerned about normal side effects, vaccine efficacy, comparing different types of vaccines, and information about each vaccine (Moderna, Pfizer, and Johnson & Johnson). These differences reveal the importance of a fine-grained taxonomy; for example, at the top category level, we would see that both groups were interested in Vaccine Safety but miss that early adopters were more concerned about normal and severe side effects, while holdouts were more concerned about eerie fears and vaccine-caused deaths.

Our taxonomy also reveals significant variability in vaccine concerns within holdouts. We observe significant differences across demographic groups; for example, holdouts from more Democrat-leaning ZCTAs were particularly concerned about FDA approval and vaccine requirements, while holdouts from more Republican-leaning ZCTAs were more concerned about eerie fears and vaccine incentives (Fig. S13). Using clustering methods, we also discover holdout profiles

| Top category | Subcategory | Description |
|---|---|---|
| **Safety** | Normal side effects | Expected side effects: sore arm, shoulder, fever, etc |
| | Severe side effects | Rare but plausible side effects, severe, potentially long-term: blood clots, myocarditis, etc |
| | Reproductive health | Concerns about fertility, breast feeding, menstruation |
| | Vaccine-caused deaths | Fear of deaths *caused* by COVID vaccine |
| | Eerie fears | Eerie and debunked fears: shedding, magnets, microchips, etc |
| | Vaccine development | History of vaccine development, fear of mRNA technology, ingredients in COVID vaccine |
| | FDA approval | FDA approval of COVID vaccines |
| **Effectiveness** | Efficacy from studies | How effective the vaccine is, how long immunity lasts, how long for vaccine to take effect |
| | Efficacy against variants | How well does vaccine work against variants (mostly Delta) |
| | Breakthrough cases | Breakthrough COVID cases, symptoms when vaccinated |
| | Natural immunity | Is natural immunity better than vaccine, do I still need vaccine |
| **Community** | Vaccine rates | Vaccine trackers, rates of vaccination over time: by state, by country, etc |
| | News on hesitancy | Reporting on vaccine hesitancy and anti-vaxxers, how to talk to vaccine hesitant |
| | Expert anti-vax | Anti-vaccine messages from scientists and doctors |
| | High-profile anti-vax | Anti-vaccine messages from high-profile figures: politicians, celebrities, etc |
| | Religious concerns | Religious concerns about the vaccine, seeking advice from religious leaders |
| **Information** | Decision-making | Pros and cons of COVID vaccine, should I get the vaccine? |
| | Comparison | Comparing Moderna vs Pfizer vs J&J, side effects, efficacy |
| | Moderna | General news on Moderna vaccine, rollout, side effects, efficacy |
| | Pfizer | General news on Pfizer vaccine, rollout, side effects, efficacy |
| | Johnson & Johnson | General news on J&J vaccine, emphasis on blood clots and efficacy |
| | Special populations | COVID-19 vaccine for special populations: autoimmune disease, rheumatoid arthritis, etc |
| | Post-vax guidelines | Guidelines after vaccination: masking, testing, quarantine |
| **Requirements** | Travel | Vaccine requirements to travel: for cruises, other countries, etc |
| | Employment | Employer vaccine mandates: healthcare, government, educators, etc |
| | Vaccine proof | Required proof of vaccination to enter places: restaurants, gyms, concert venues, etc |
| | Exemption | Seeking exemption on vaccine requirements, religious or medical |
| | Fake vaccine proof | Seeking fake proof of vaccination |
| | Anti-mandate | States banning mandates, lawsuits against employer mandates |
| **Incentives** | Vaccine incentives | Vaccine incentives: lotteries, gift cards, free groceries, giveaways, etc |
| **Availability** | Locations | Where to get COVID vaccine (some missed vaccine intent URLs): CVS, Walgreens, etc |
| | Children | Are COVID vaccines for children available / recommended |
| | Boosters | Are boosters available / recommended |
| **Other** | New / non-US vaccines | Other COVID vaccines: Novavax, Astrazeneca, Sinovax |
| | Non-COVID vaccines | Non-COVID vaccines: flu, MMR, varicella, meningitis, etc |
| | Pet vaccines | Vaccines for pets, mostly dogs and cats |

**Fig. 3 | Taxonomy of vaccine concerns.** Our taxonomy consists of 8 top categories and 36 subcategories.

directly from their expressed vaccine concerns. Four distinct profiles emerge (Fig. S16): one represents the stereotypical holdout (interested in vaccine misinformation and anti-vaccine messages), one focused on government policies (vaccine requirements and incentives), one engaging with decision-making (analyzing pros and cons of receiving a COVID-19 vaccine), and one seeking information about specific vaccine brands and side effects. These profiles illustrate different types of holdouts, who vary in their openness to the vaccine and their key

concerns, which implies that policymakers need to go beyond one-size-fits-all solutions to address vaccine hesitancy.

**Holdouts appear like early adopters when seeking the vaccine**
In our final analysis, we exploit the fact that all of our vaccine holdouts eventually expressed vaccine intent to explore how vaccine concerns change as an individual converts from holdout to adopter. From July to August 2021, we analyze how holdouts' vaccine concerns change in the

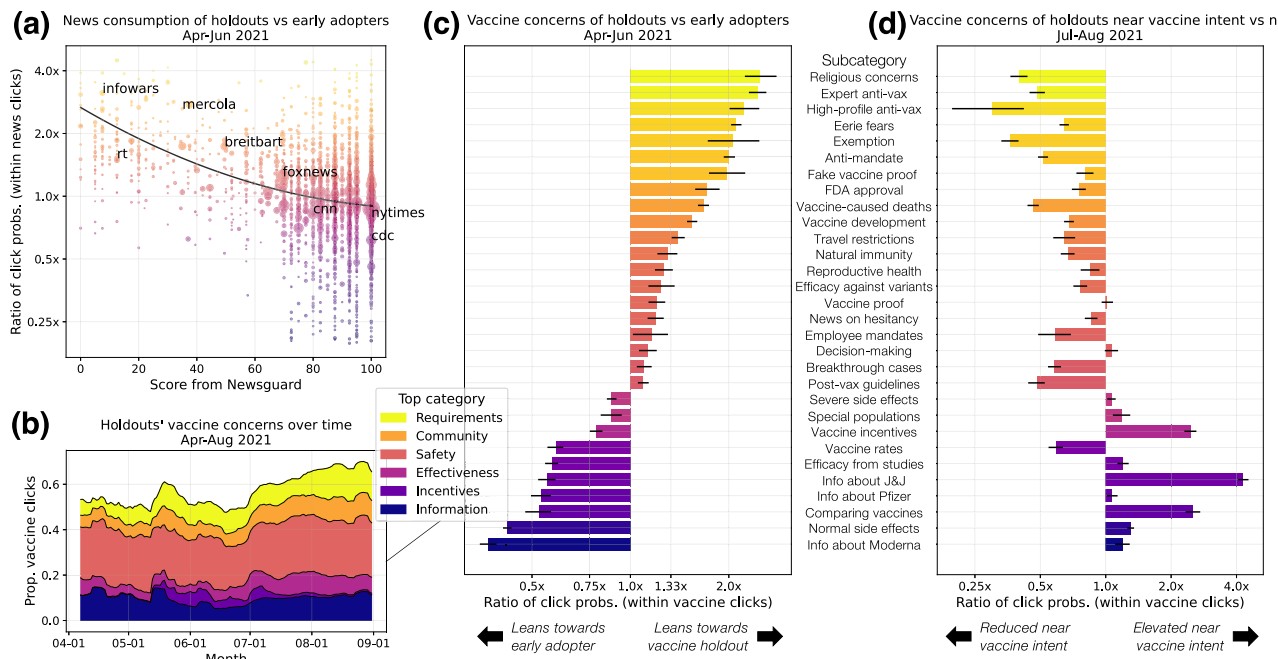

**Fig. 4 | Vaccine concerns and news consumption.** In all subfigures, news/categories are colored from yellow to dark purple to represent most holdout-leaning to most early adopter-leaning. **a** The lower the trust rating from Newsguard, the likelier it is that vaccine holdouts click on the news domain, relative to early adopters. Each dot represents a news domain, and we include labels for a few notable domains, such as infowars.com or cdc.gov (with their extensions removed for brevity). **b** Holdouts' top category concerns include Vaccine Safety, Requirements, and Information, with varying proportions over time. **c** Comparing holdouts vs. early adopters' relative probabilities of clicking on each subcategory (from April to June 2021) reveals each group's distinctive concerns. Exact values are reported in Table S6. **d** Near when holdouts express vaccine intent (± 3 days) in July and August 2021, their concerns become much more like the concerns of early adopters, with a few important differences. Exact values are reported in Table S7. In (**c**–**d**) error bars indicate 95% CIs computed over 1000 bootstrapped samples ("Analyses of news consumption and vaccine concerns").

small window (± 3 days) surrounding their expressed vaccine intent, compared to their typical concerns outside of that window. We find that in those windows, holdouts' vaccine concerns nearly reverse, such that they look much more like early adopters than their typical selves (Fig. 4d nearly reverses 4c). During this time, holdouts become far more interested in the Johnson & Johnson vaccine, comparing different vaccines, and vaccine incentives, and less interested in anti-vaccine messages and vaccine fears (Table S7). Notably, not all early adopter-leaning concerns reverse as dramatically; for example, even while expressing vaccine intent, holdouts remain less interested in the Pfizer and Moderna vaccines, which may reflect how vaccine hesitant individuals were quicker to accept the one-shot Johnson & Johnson vaccine, instead of the two-shot mRNA vaccines[61,62]. Furthermore, there are some early adopter-leaning concerns that holdouts do not pick up on during this time, such as interest in regional vaccine rates. We hypothesize that these concerns are more reflective of an early adopter personality rather than of concerns that become immediately relevant when seeking the vaccine, such as comparing different vaccines. Finally, we study the problem of trying to predict when a holdout will show vaccine intent, and we show that vaccine concerns and news consumption from the past week both significantly improve predictive performance. Thus, search logs reveal which holdouts are closer to conversion, which—if deployed with proper privacy protections—can help to guide budgeted interventions from policymakers.

## Discussion

We have demonstrated how large-scale search logs and machine learning can be leveraged for fine-grained, real-time monitoring of vaccine seeking and vaccine concerns. Still, there are limitations to our approach: for example, while we can achieve finer granularity than existing data, we still miss within-ZCTA heterogeneity in vaccine intent. Furthermore, our efforts to minimize bias in the estimated vaccine

intent rates are substantial but imperfect (e.g., we can only approximate the true and false positive rates of our classifier). We also assume in this work that vaccine intent can be detected through single queries or clicks, but more sophisticated models could incorporate entire search sessions or browsing data beyond search. However, in favor of simplicity and considerations of privacy, we label vaccine intent at the query and click-level. We are mindful throughout this work of the need to balance privacy and social benefits when using potentially sensitive user data. For this reason, we only link as much as we need at the individual-level (e.g., without demographic attributes or browsing data) and only report aggregated results.

Despite these limitations, our resources demonstrate strong agreement with existing data and enable analyses that have not been available before. Our vaccine intent classifier achieves high correlations with vaccination rates reported by the CDC, but it also allows us to estimate vaccine rates down to the ZCTA-level. This spatial granularity supports more precise analyses and interventions; for example, the finer-grained estimates can help public health officials to identify under-vaccinated communities, informing where to place vaccine sites or whom to prioritize in online or real-world outreach programs. Our vaccine intent signals also precede and improve prediction of daily vaccinations, showing promise as information that policymakers could use to preempt vaccine demand. Finally, the real-time nature of search signals opens up opportunities to assess the effects of interventions, such as determining if a new public service announcement is responsible for a proximal rise in vaccine interest, or understanding how people learned about a new vaccination location through search.

While our vaccine intent classifier can be harnessed to provide insights about where to intervene and for whom, our taxonomy and analyses of vaccine concerns inform how to intervene. Search logs offer a glimpse into individuals' genuine interests and exposure to

content, without depending on self-reported survey data or user-generated content on social media. By examining holdouts' online news consumption and specific vaccine concerns, our findings could help shape messaging strategies in vaccination campaigns. For example, we show that even within holdouts, their key vaccine concerns vary significantly across demographics and profiles, requiring tailored messages to address vaccine hesitancy. Furthermore, we provide novel insights about the temporal dynamics of vaccine concerns and vaccine seeking. Our observation that holdouts resemble early adopters when they eventually seek vaccination indicates that individuals might follow similar paths towards vaccine acceptance. Future work could try to model these trajectories, such as identifying key influences (e.g., vaccine mandates) that move individuals along the path toward acceptance. Finally, our finding that recent vaccine concerns and news consumption predict the timing of vaccine intent suggests that search logs can help to allocate limited resources for interventions (e.g., online advertising campaigns), by identifying individuals who are closer to conversion.

Our methods demonstrate the potential of large-scale computing platforms, utilized by millions nationwide, to offer valuable anonymized public health signals that might be otherwise challenging to acquire. Particularly given the decentralized nature of the US public health system, depending on a patchwork of state and county public health agencies, insights based on data drawn from nationwide computing platforms can play a valuable integrative role. While we focus in this work on the COVID-19 vaccine in the US, our approach is not specific to this vaccine or location, and we hope that future work can extend our methods to study vaccination behaviors in other countries and languages, as well as examining the deployment and adoption of other vaccines.

To facilitate policy impact and future research, we have released our vaccine intent estimates and our taxonomy of vaccine concerns. We hope that these resources will be useful to policymakers and researchers so that they can conduct detailed analyses of COVID-19 vaccine rates, such as evaluating the efficacy of vaccine distribution plans and studying disparities in vaccination rates. The taxonomy can also be employed widely in web and social media research, since it includes tens of thousands of URLs and their categorization. For example, it can be used to study how certain classes of URLs (e.g., eerie fears) are disseminated on social media or surfaced by search engines. Finally, we note that our graph-based machine learning techniques for intent detection on search are applicable beyond vaccines, and could be applied to precisely detect other intents of interest, such as registering to vote or filing for stimulus checks. More broadly, we hope that our work can serve as a roadmap for researchers of how to derive rigorous behavioral and health insights from search logs, including how to precisely detect user intents and interests, evaluate and correct for bias in estimates, validate predictions against external data, and release resources to promote reproducibility, transparency, and future work.

## Methods

The Methods section is structured as follows: In "Datasets", we discuss the datasets we use, including Bing search logs, CDC vaccination rates, and data from the US Census. In "Vaccine intent classifier", we describe our methods to develop a vaccine intent classifier. In "Estimating vaccine intent rates and correcting for bias", we discuss how we apply our classifier to estimate regional vaccine intent rates, correct for and evaluate bias in our estimates, and compare to external data from the CDC and Google. In "Taxonomy of vaccine concerns on search", we describe our methods to construct our taxonomy of vaccine concerns. Finally, in "Main analyses", we provide methodological details about our main analyses. Supplementary analyses, figures, and tables are provided in the Supplementary Information (SI).

## Datasets

**Bing search logs.** Our work leverages billions of anonymized search logs from Bing. Bing is the second largest search engine worldwide and in the US, with a US market share of around 6% on all platforms and around 11% on desktop[63]. Despite having non-uniform coverage across the US, Bing has enough penetration across the country that we can estimate representative samples after applying inverse proportional weighting ("Coverage-corrected vaccine intent rates"). The Bing data we use consist of individual queries made by users, where for each query, we have information including the text of the query, an anonymized ID of the user, the timestamp, the estimated geolocation (ZIP code, county, and state), and the set of URLs clicked on, if any. Since our work is motivated by insufficient vaccine data and vaccine concerns in the US, we limit our study to search logs in the US market. However, the methods we introduce could be extended to study vaccination rates and vaccine concerns in other languages and countries. We apply our vaccine intent classifier ("Vaccine intent classifier") to Bing search logs from February 1 – August 31, 2021. February 2021 was the earliest that we could study following data protection guidelines, which allow us to store and analyze search logs up to 18 months in the past. We end in August 2021, since the FDA approved booster shots in September and our method is not designed to disambiguate between vaccine seeking for the primary series versus boosters. Our work was approved by the Bing product team, in addition to other privacy officers at Microsoft (see Data ethics below).

**US Census.** We estimate vaccine intent rates at the level of ZIP code tabulation areas (ZCTAs)[64], since they are the smallest Census-tracked unit to which we can reliably map Bing queries and users ("Coverage-corrected vaccine intent rates"). To characterize demographic trends in vaccine intent, we use ZCTA data from the US Census' 2020 5 year American Community Survey[65]. The demographic variables we use are total population size, percent female, percent of different age groups (e.g., under 18, over 65), percent of different races/ethnicities (White, Black, Asian, Hispanic), percent with Bachelor's degree or higher, and population per square meter (in log scale), which divides the population size by the ZCTA's land area[66]. To create map visualizations (e.g., Fig. 2a), we also use the 2020 ZCTA, county, and state shapefiles provided by the US Census[67].

**Reported vaccination rates.** To evaluate our vaccine intent classifier, we compare it to reported vaccination rates in the US ("Comparison to reported vaccination rates"). First, we use data from the Centers for Disease Control and Prevention (CDC), who provide daily cumulative vaccination rates at the levels of states[68] and counties[69]. They provide different measures, including the total number and percentage of population who have received at least one dose, completed a primary series, received a booster shot, received a second booster shot, and so on. The CDC does not provide ZCTA-level vaccination rates, but they are provided by some states, such as New York and California. We use two ZCTA-level datasets, one from the California Department of Public Health[70] and one from 16 large US cities, which were compiled by the Big Cities Health Coalition's COVID-19 Health Inequities in Cities Dashboard[71] and made available by Bilal et al.[72].

**Elections data.** To capture political lean per region, we use county-level data from the 2020 US presidential election, which we purchased online[73]. In our analyses, we use "percent Republican" as a variable, i.e., the percentage of overall votes cast in the county that went to the Republican nominee Donald Trump.

**Newsguard data.** We use data from Newsguard to label the trustworthiness of different news sites. Newsguard assigns numerical trust scores to news sites based on nine journalistic criteria, such as how

often the site publishes false content, how responsibly it collects and presents information, how it handles the difference between news and opinion, and how transparent it is about its funding[55]. News sites with scores above 60 are considered Trusted and below 60 are considered Untrusted. They also categorize other sites as Satire or Platform, but these sites are not given numerical scores since the criteria are not as relevant. In our analysis, we focus on the Trusted and Untrusted sites with numerical scores ("Analyses of news consumption and vaccine concerns").

**Google search trends**. To evaluate the representativeness of Bing search trends, we compare them to Google search trends ("Comparison to Google search trends"). Google allows individuals to view aggregated, normalized search trends for any query with enough users[74]. The trends for that query over time and across subregions (e.g., US states) are then available for download.

**Data ethics**. Our work was approved by both the Microsoft IRB office, and by an internal privacy review process which included privacy officers from both Microsoft Research and the Bing product team. Together, we worked to ensure that our use of Bing search logs was consistent with Bing's privacy policy (which explicitly lists research as a possible use of search data), and with relevant company best-practices. When we use search logs, we are mindful of the need to balance privacy and social benefits when using potentially sensitive user data. While we study individual search logs, since we need to be able to link individual vaccine intent to search interests, those sessions are assembled using only anonymous user identifiers, which are disassociated from any specific user accounts or user profiles, and cannot be linked to any other Microsoft products. Likewise, in this anonymous view of the logs, location and demographic data were limited to ZIP code-level accuracy. Finally, we are careful to only report results aggregated over thousands of individuals. Aside from Bing search logs, all of the data sources we use here are publicly available and aggregated over many individuals.

## Vaccine intent classifier

We develop a vaccine intent classifier to detect users who are expressing vaccine intent, i.e., seeking the COVID-19 vaccine with web search. As we discuss in the main text, we can use regular expressions to identify queries expressing vaccine intent, such as [where can i get a covid vaccine]. However, intent can also be clarified through clicks on search results[40], such as clicking on the CVS registration page for the COVID-19 vaccine. The challenge with URLs is that we cannot easily define regular expressions to identify the ones expressing vaccine intent. Instead, our approach is to construct query-click graphs, then to use a combination of graph-based machine learning techniques and manual annotation to identify a large set of vaccine intent URLs. Our method consists of three steps: use personalized PageRank to propagate labels from queries to URLs, so that we can generate a set of URL candidates for manual annotation ("Personalized PageRank for URL candidates"); present the URL candidates to external annotators on Amazon Mechanical Turk to label as vaccine intent or not ("Annotation on Amazon Mechanical Turk"); use the labels from the previous step to train graph neural networks so that we can further expand our set of vaccine intent URLs ("Graph neural networks for URL expansion").

### Personalized PageRank for URL candidates
**Constructing query-click graphs**. Our first goal is to construct query-click graphs that broadly cover searches related to COVID-19 and the vaccine. First, we collect all queries that contain any word from the following list of keywords: {"covid", "corona", "pandemic", "cov19", "virus", "variant", "vaccin", "vacin", "vax", "dose", "shot", "booster", "rollout", "roll out", "fda", "cdc", "johnson", "jj", "janssen", "pfizer", "phizer", "biontech", "moderna", "astrazeneca", "mrna"}. We

constructed this list by starting with a smaller core set, {"covid", "vaccine", "vaccin", "booster"}, collected queries that included any of those words, and looked through the top 1000 words that appeared in those queries. Then, we also collect all queries and clicks that co-occurred in a search session with any of these queries. Using co-occurrence allows us to capture vaccine-related queries and URLs that do not include the keywords, such as California's vaccine scheduling page, myturn.ca.gov. We construct a query-click graph from all of these queries and clicks, with queries and URLs as nodes. Our graph consists of two types of edges: first, an edge from query A to query B represents that A preceded B in a search session; second, an edge from a query to a URL represents that searching that query led to a click on that URL. In both cases, the edge weight indicates the number of times this relationship appears in our data.

Since URLs may appear or disappear over time, we collect queries and clicks from two spread-out months in our study period, April 1-30, 2021 and August 1-31, 2021. We construct query-click graphs *separately* for every US state, since we find that the classifier performs better across states when we build different graphs and models per state ("Bias in vaccine intent classifier"). We also perform minor pre-processing at this step: we lower-case queries, drop queries that are implausibly long (over 100 characters), and drop clicks that are not on URLs (e.g., "javascript:void").

**Personalized PageRank from query seed set (S-PPR)**. We define vaccine intent queries as those that are *unambiguously* seeking the COVID-19 vaccine with web search. To be included, the query must include both a COVID-19 term ("covid" or "coronavirus") *and* a vaccine term ("vaccin", "vacin", "vax", "dose", "shot", "booster", "johnson", "pfizer", or "moderna"). In addition, the query must satisfy at least one of the following criteria: (1) matching some variant of "find me a COVID-19 vaccine", (2) containing appointment-related words (e.g., "appointment", "sign up") or location-seeking words (e.g., "near me", "where can i get"), (3) containing a pharmacy name. We try to capture a representative list of pharmacy names by including almost all pharmacies that provided the COVID-19 vaccine through the Federal Retail Pharmacy Program for COVID-19 Vaccination, which includes 21 pharmacy partners and their many subsidiaries[75]. Based on an inspection of the queries containing each pharmacy name, we drop a few names that are ambiguous (e.g., United is a pharmacy, but "united" can be confused with United States or United Airlines).

To identify URL candidates for vaccine intent, we use personalized PageRank. Personalized PageRank[43] is a common technique for seed expansion, where a set of seed nodes in a graph are identified as members of a community, and one wishes to expand from that set to identify more community members[44]. In our case, the vaccine intent queries act as our seed set, and our goal is to spread the influence from the seed set over the rest of the query-click graph. For a given seed set $S$, personalized PageRank derives a score for each node in the graph that represents the probability of landing on that node when running random walks from $S$. The hyperparameter $\alpha$ controls the lengths of the random walks by defining the probability of continuing the random walk versus teleporting back to the seed set. In our work, we use the default $\alpha = 0.85$. Personalized PageRank naturally trades off between favoring nodes that are closer to the seed set (if random walks are shorter) and nodes that are central in the network (if random walks are longer). These are also the two desiderata of the URL candidates we hope to find: they should be close to the vaccine intent queries in the graph and, to achieve high utility from labeling, they should be central and high-degree (i.e., we would not want to "spend" a label on a URL that is rarely clicked on).

Thus, we run personalized PageRank from the seed set of vaccine intent queries (S-PPR) to derive scores for all URLs in each query-click graph. S-PPR also provided scores for all queries in the graph, but we found that our seed set was quite comprehensive in identifying

unambiguous queries. The top-ranked queries that were not in the seed set tended to be location-specific, such as [covid vaccine new york], which were suggestive of vaccine intent, but we decided were not unambiguous enough (e.g., they could be seeking vaccine locations in New York, but could also be seeking information about vaccine eligibility or requirements in New York).

**Selecting URL candidates.** In this step, we select the URL candidates that external annotators will manually label for vaccine intent. First, we filter out URLs that do not begin with "http", which leaves out URLs that are ads, internal links to other Microsoft verticals (e.g., News, Videos), and telephone numbers. Then, we order the remaining URLs in each state according to their S-PPR scores, from highest to lowest. We keep the *union* over states of their top 100 URLs as our set of URL candidates, resulting in 2483 candidates. The number of URLs we have from taking the union over states is much lower than the number of states multiplied by 100, since there is overlap between states. For example, the COVID-19 vaccine page for Walgreens is one of the most common URLs, appearing in the top 100 for all 50 states with an average ranking of 5.46 (where 0 indicates top-ranked). However, there is also substantial heterogeneity in top URLs across states, reflecting state-specific vaccine programs and policies (Table S1). By constructing separate graphs and running S-PPR per state, our approach is uniquely able to capture this state-specific heterogeneity. In "Bias in vaccine intent classifier", we show how alternative approaches that use a combined graph for multiple states severely hurt performance for small states.

Before presenting our URL candidates to annotators, we perform additional post-processing on the candidates in the union set. First, we identify highly similar URL patterns that appear (such as the CVS store locator for the COVID-19 vaccine, which always begins with www.cvs.com/store-locator/cvs-pharmacy-locations/covid-vaccine/), and only keep up to 5 URLs per pattern, so that our annotations are not overly repetitive. This process reduces our set to 2222 URLs. Second, a limitation of our annotations is that we collected these annotations approximately a year after the end of our study period (August 2021). From our search data, we know the original URL that they clicked on, but we do not have the original contents of the page. To identify URLs that redirect to different pages now, for each URL candidate, we compute the normalized edit distance between the requested URL and the URL it redirects to by taking the Levenshtein distance divided by the length of the requested URL. We keep URLs with a normalized edit distance less than or equal to 0.2, which keeps around 80% of URLs. In other cases, the contents of the page may have changed while the URL remained the same. However, we/the annotators are able to see the original URL that was clicked on, and URLs are often informative, since they are often a hyphenated version of the page title or closely related to it. Furthermore, many of the URLs labeled were news articles, which do not tend to change over time. Finally, our classifier also relies heavily on queries, and we can see the original query. So, content shift on the page is a limitation, but this concern is mitigated by redirected URLs being removed, the original URL being available, limited time between the study period and annotation period, news articles rarely changing over time, and the classifier also relying on queries.

**Annotation on Amazon Mechanical Turk**
**Gathering annotations.** In this step, we present our URL candidates to annotators on Amazon Mechanical Turk (AMT). Our task instructs them to click on the presented URL and to answer based on what they see. The first question we ask is, "Given that a person clicked on this page during a search session, how sure are you that this person is seeking to get the COVID-19 vaccine (any dose or booster)?" (Fig. S1a). We provide options from Highly Likely to Unlikely, as well as Missing Page. If the annotator selected Likely, Ambiguous, or Unlikely, we ask them to indicate what other intention(s) the person might have, such

as seeking information about vaccine safety or COVID-19 testing (Fig. S1b). While we use the answers to the first question to construct our vaccine intent labels, we include the second question to encourage annotators to think broadly about vaccine-related searches, so that they would only label positively for vaccine intent if the URL seemed unambiguous. To validate our vaccine intent queries, we also present a sample of queries to annotators. To capture a diverse sample, we use the union over the top 5 and bottom 5 vaccine intent queries per state, after filtering out queries that were issued by fewer than 50 users (for privacy reasons) and sorting the remaining ones by their S-PPR scores. This results in 227 vaccine intent queries to label. In the query version of our task, we ask very similar questions to those shown in Fig. S1 but replace language about clicking on the URL with issuing the query.

To make sure our questions were clear, we conducted two internal user studies, first with the authors doing a small pilot run, then with recruited colleagues at Microsoft doing a larger pilot run. Our pilot studies allowed us to test the design of the questions, but our final vaccine intent labels were entirely based on the AMT labels that we received. Our pilots also allowed us to estimate that each task (labeling a single URL or query) would take around 30 seconds. We set the compensation on AMT to $0.15 per task, which corresponds to an hourly rate of around $18. Our AMT task was also approved by the Microsoft IRB Office, in a separate application from our approved analysis of Bing search logs, and we included a consent form in our instructions that annotators were required to read and sign before starting the task.

**Annotation results.** For each URL, we first present it to three annotators. If all three give it a positive label (i.e., Highly Likely or Likely), then we label this URL as vaccine intent. If two give it a positive label and one does not, we consider this a "non-consensus" URL, and we assign it to one more annotator. If that annotator gives it a positive label, then we also label this URL as vaccine intent. In other words, we require vaccine intent URLs to receive three positive annotations. With this relatively strict bar, we still find that a large majority (86%) of our URL candidates are labeled as vaccine intent. We also find a very high positive rate among the vaccine intent queries that we tested. Using the same annotation process and requirement of three positive labels, we find that 96% of the vaccine intent queries we test are labeled as true vaccine intent. The ones that are not seem to be mislabeled due to noise and our high bar for inclusion, since on inspection, they do seem to unambiguously communicate vaccine intent (e.g., [covid vaccines walgreens]).

Furthermore, we observe a clear relationship between S-PPR rank and the percentage labeled as vaccine intent: for example, around 90% of URLs from ranks 0 to 20, around 81% of URLs from ranks 40–60, and around 71% of URLs from ranks 80 to 100 (Fig. S2). The agreement between S-PPR rank and our acquired annotations both support the ability of S-PPR to predict vaccine intent remarkably well and provides evidence that the annotations are meaningful. We also calculate interannotator agreement on this task. As before, we bucket the annotations into positive (Highly Likely or Likely) and negative (Ambiguous or Unlikely) labels. Then, we compute the observed agreement $P_0$ between annotators over all URLs and pairs of annotators. Let $pos(i)$ and $neg(i)$ represent the number of positive and negative labels received for URL $i$, respectively; note that $pos(i) + neg(i) \leq 4$, since we have at most four annotators per URL.

$$P_0 := \frac{\sum_i pos(i) \cdot (pos(i) - 1) + neg(i) \cdot (neg(i) - 1)}{\sum_i (pos(i) + neg(i))(pos(i) + neg(i) - 1)}. \quad (1)$$

We find $P_0 = 73\%$, which is substantially higher than what we would expect to see by chance (50%), which results in a Randolph's $\kappa$[76] of 0.46. We use Randolph's $\kappa$, which is appropriate for our setting since it allows for multiple, *flexible* number of raters per data point, while

Fleiss' $\kappa$ assumes fixed marginals (i.e., equal number of raters). Finally, for additional quality control, we review the URLs labeled as vaccine intent and find that the majority of them seem correct. We remove a small number of URLs that still seem slightly ambiguous, e.g., the COVID-19 page for a jurisdiction that may not be vaccine-specific. In our review, we also find common URL patterns, such as COVID-19 vaccine store locators for CVS or Walmart. We define regular expressions matching these patterns so that we can detect these types of URLs too, even if they are not in our predefined vaccine intent set.

**Graph neural networks for URL expansion.** Since manual annotation is expensive, we wish to augment our manual efforts by training machine learning models on the AMT labels, then use the models to expand our set of vaccine intent URLs. We formulate this problem as semi-supervised node classification on a graph, since the URLs are nodes in the query-click graph and we are trying to predict whether a URL indicates vaccine intent or not, given labels for a subset of URLs. To solve this problem, we use graph neural networks[45] (GNNs), which are a powerful class of machine learning models for graph-structured data that naturally incorporate graph structure and node features into prediction.

**Training GNNs to predict vaccine intent.** Our model consists of two character-level convolutions (CNN), followed by three graph convolutions (GCN), followed by a final linear layer and sigmoid activation that produces the model prediction as a probability between 0 and 1. We use the character-level CNN to capture textual information in the queries and URLs, since text can be informative for this problem (e.g., the appearance of "vaccine" or "vax"). Character-level representations are more natural for URLs, which often join words through hyphens or concatenation, or use abbreviations or truncated words. Character-level representations also allow us to account for typos in queries. The graph convolutions then allow us to learn representations of URLs that draw from the representations of their neighboring queries, which draw from the representations of their neighboring URLs, and so on. In this way, we can capture "similar" URLs in embedding space (similar in terms of both text and graph structure), which allows us to learn embeddings that are predictive of vaccine intent.

Given the query-click graph for a state, we label a URL node as 1 if the URL was labeled as vaccine intent from AMT (following the inclusion criteria described in the previous section) or if it matches one of the regular expressions we identified for vaccine intent URLs. We label it as 0 if at least two AMT annotators gave the URL negative labels (Ambiguous or Unlikely) or if it matches a regular expression we identified as *not* vaccine intent, such as general store locators for pharmacies. To train and test our model, we randomly split the URL labels into a train set (60%), validation set (15%), and test set (25%). We train the model on the train set, iteratively updating model parameters with gradient descent on the train loss (cross-entropy loss) and evaluating its loss on the validation set. We continue training until the model's validation loss is no longer improving. Finally, we evaluate the model's performance on the held-out test set using area under the receiver operating characteristic curve (AUC), a standard metric in machine learning. In "Bias in vaccine intent classifier", we additionally evaluate the model's true and false positive rates, which are central to evaluating the model's bias ("Decomposition of bias").

However, some states have much smaller graphs, and therefore, fewer positive and negative labels. For example, for the state of Wyoming, we only have 245 positive and 276 negative URLs. We find that with such few labels, the model cannot adequately learn how to predict vaccine intent, with AUCs far below those of large states (Table S2). Additionally, as we show below, we find that joining state graphs into one combined graphs also results in worse performance for smaller states, since larger states' labels and query-click patterns dominate. Instead, our key insight is that we can retain state-level

graphs and models, but we *pre-train* the model for smaller states on S-PPR rankings ("Personalized PageRank for URL candidates"), which we have for many more URLs than we have labels for. Our intuition is that S-PPR already performed remarkably well at predicting vaccine intent, as we showed in our annotation results ("Annotation on Amazon Mechanical Turk"). Furthermore, S-PPR rankings do not require any additional manual labels; we derive them entirely from our initial vaccine intent queries, which were automatically labeled using regular expressions. In practice, before training the model on the URL labels from AMT and regular expressions, we train the model to predict the URLs' S-PPR rankings that we derived in Step 1. Since S-PPR rankings become less meaningful in the long tail of URLs, we focus on the top $K = \max(1000, q_{max})$ S-PPR rankings, where $q_{max}$ is the maximum rank (where lower rank corresponds to higher S-PPR score) of the last seed set query. This pre-training encourages the model to learn URL representations that are predictive of S-PPR rankings, which we find help substantially with the ultimate task of predicting vaccine intent.

**Evaluating GNN performance.** We evaluate model performance on the held-out test set by computing its AUC, which captures how well the model trades off between its true positive rate and false positive rate. Furthermore, to account for randomness from model training and data splitting, we run 10 random trials for every model/state, where in each trial, we re-split the URL labels into train, validation, and test sets, retrain the model on the train set (stopping based on the validation loss), and re-evaluate the model's final performance on the test set.

First, we select six representative states, chosen to vary in graph size and US region, to test the effect of pre-training on S-PPR rankings. We find that pre-training significantly improves performance for the smaller states; for example, the mean AUC for Wyoming increases from $0.74 - 0.95$ (Table S2, Fig. S3). Specifically, due to the low number of URL labels for smaller states, we observe great variance in the model's performance if we do not pre-train the model, leading to some trials that perform well and some that perform poorly. Performance becomes far more stable for smaller states after we incorporate the pre-training objective. We find that pre-training seems unnecessary for the larger states, such as Connecticut and Tennessee, where we are already achieving high AUCs above 0.98. So, we set a generous cutoff of 5,000,000 nodes (still larger than the graph size for Connecticut) and we pre-train all states with fewer than 5,000,000 nodes in our data, of which there are 26. After incorporating pre-training for these smaller states, we are able to achieve AUCs above 0.90 for all 50 states and above 0.95 for 45 states (Fig. S4). These results demonstrate that our GNNs are able to accurately predict vaccine intent labels in all 50 states, which is essential as we use our GNNs to discover new vaccine intent URLs.

In the SI, we conduct a supplemental analysis showing that, before providing the GNN with *any* URL labels, the GNN pre-trained on S-PPR rankings already outperforms S-PPR at predicting URL labels (Fig. S17), by 10–15 points in AUC. These results show that, due to the expressive power of the GNN (with character-level CNN) and the predictive power of S-PPR from a well-designed seed set, we can achieve decent performance without any labels at all. These methods, which could be explored more deeply in future work, may be useful in a zero-shot context, allowing lightweight, effective prediction before acquiring any labels.

**Discovering new vaccine intent URLs.** Finally, we use our trained GNNs to identify new vaccine intent URLs. We apply our GNNs to predict scores for all unlabeled URLs within the top $K$ URLs according to S-PPR ranking (again, with $K = \max(1000, q_{max})$). However, in order to decide which new URLs to include as vaccine intent, we need to determine a score threshold. Our goal is to set the threshold such that any URL that scores above it is very likely to truly be vaccine intent (i.e.,

we want to have high precision). Borrowing the idea of "spies" from positive-unlabeled learning[77], our idea is to use the held-out positive URLs in the test set to determine where to set the threshold. We consider two thresholds: (1) $t_{\text{med}}$, the median score of the held-out positive URLs, and (2) $t_{\text{prec}}$, the minimum threshold required to achieve precision of at least 0.9 on the held-out test set. Then, we only include URLs that pass both thresholds in at least 6 out of the 10 random trials (where, as described before, we reshuffle the data and retrain the model per trial).

Our method is similar to common "big data" approaches that, due to the scale of unlabeled data, seek to manually annotate a subset of data, train machine learning models to accurately predict those labels, then use those models to label the rest of the data[78–81]. We extend this approach with special attention to the classification threshold, setting it high so that we can ensure high precision among the new URLs that we discover. Even with this high threshold, we discover around 11,400 new URLs, increasing our number of vaccine intent URLs by 10 x. In Table S3, we provide a uniform random sample of the URLs that our GNNs discovered. The majority of them seem to express vaccine intent, with several news stories about new vaccine clinics and information about vaccine appointments. In the following section, we also evaluate the impact of adding these URLs discovered by GNNs on our ability to estimate regional vaccine intent rates. We find that the new URLs not only increase our coverage of vaccine intent users by 1.5 x but also further improve our agreement with reported vaccination rates from the CDC (Table 1).

### Estimating vaccine intent rates and correcting for bias

In this section, we discuss how we use our classifier to estimate regional rates of vaccine intent and how we correct for and evaluate sources of bias in our estimates.

**Decomposition of bias.** For a given individual, let $v \in \{0, 1\}$ indicate whether they actually had vaccine intent (up to a certain time) and $\hat{v} \in \{0,1\}$ indicate whether our classifier labels them as having vaccine intent. Furthermore, let $r$ represent the individual's home region, such as their state or county. We would like to estimate the regional vaccine intent rate, $\Pr(v|r)$, but we do not have access to $v$, only to $\hat{v}$. To understand how simply using $\hat{v}$ in place of $v$ may bias our estimates, let us relate $\Pr(\hat{v}|r)$ to $\Pr(v|r)$. First, we introduce another variable $b$, which represents whether the individual is a Bing user. Note that $\hat{v} = 1$ implies that $b = 1$, since our classifier can only identify vaccine intent from users who appear in Bing search logs. With these variables, we have

$$
\Pr(\hat{v}=1|r) = \underbrace{\Pr(b=1|r)}_{\text{Bing coverage of } r}
$$
$$
\left[ \Pr(v=1|r) \underbrace{\Pr(\hat{v}=1|b=1, v=1, r)}_{\text{Classifier TPR for } r} + \Pr(v=0|r) \underbrace{\Pr(\hat{v}=1|b=1, v=0, r)}_{\text{Classifier FPR for } r} \right],
$$

(2)

where TPR and FPR are the true and false positive rates, respectively. $\Pr(b=1|r)$ represents the probability that an individual from region $r$ is a Bing user, i.e., the Bing coverage of $r$. Incorporating $b$, $v$, and $r$ into $\Pr(\hat{v}|b, v, r)$ reflects all of the factors that affect whether the classifier predicts vaccine intent. As discussed, if the user is not a Bing user ($b = 0$), then the probability is 0, so we only consider the $b = 1$ case. If $v = 1$, predicting $\hat{v} = 1$ would be a true positive; if $v = 0$, it would be a false positive. Conditioning $\hat{v}$ on region $r$ reflects the possibility that individuals from different regions may express vaccine intent differently and the classifier may be more prone to true or false positives for different regions. Finally, we make the assumption here that $b \perp v | r$; that is, conditioned on the individual's region, being a Bing user and having vaccine intent are independent. This misses potential within-region

heterogeneity, but to mitigate this in practice, we focus on fine-grained regions (ZIP code tabulation areas, "Coverage-corrected vaccine intent rates").

Based on this decomposition, we can see that if Bing coverage, TPR, and FPR are uniform across regions, then $\Pr(\hat{v}|r)$ will simply be a linear function of $\Pr(v|r)$. Unfortunately, we know that Bing coverage is not uniform. However, we observe $b = 1$ and can assign users to regions, so we can estimate Bing coverage per region and correct by inverse coverage. Thus, our estimate corresponds to a coverage-corrected predicted vaccine intent rate, $\tilde{p}(v, r) = \frac{\Pr(\hat{v}=1|r)}{\Pr(b=1|r)}$. If we refer to the true vaccine intent rate as $p(v, r)$, then we can see that $\tilde{p}(v, r)$ is a linear function of $p(v, r)$ when TPR and FPR are uniform:

$$
\begin{aligned}
\frac{\Pr(\hat{v}=1|r)}{\Pr(b=1|r)} &= \Pr(v=1|r)\text{TPR} + (1 - \Pr(v=1|r))\text{FPR} \\
\tilde{p}(v, r) &= \text{FPR} + (\text{TPR} - \text{FPR})p(v, r).
\end{aligned}
$$

(3)

Furthermore, if FPR is low, then $\tilde{p}(v, r)$ is approximately proportional to $p(v, r)$. Thus, our first two strategies for addressing bias in our estimates are:

1. Estimate Bing coverage per region and weight by inverse coverage ("Coverage-corrected vaccine intent rates"),
2. Evaluate whether we observe similar TPRs and FPRs across regions and whether FPRs are close to 0 ("Bias in vaccine intent classifier").

These efforts are our first two lines of defense against bias. After this, we can furthermore compare our final vaccine intent estimates to established data sources, such as the CDC's reported vaccination rates ("Comparison to reported vaccination rates") and Google search trends ("Comparison to Google search trends").

#### Coverage-corrected vaccine intent rates

**Estimating Bing coverage.** Our goal here is to estimate $\Pr(b=1|r)$, the probability that an individual from region $r$ is a Bing user. We focus on ZIP Code Tabulation Areas (ZCTAs) as our fine-grained notion of regions; for example, there are ~10x more ZCTAs in the US than counties. ZIP codes are the most granular geographic area that we can assign Bing users to, since we have, for most Bing queries, a record of which ZIP code the query came from. We focus on ZCTAs, which are "generalized areal representations" of ZIP codes, since they are a unit that the Census tracks and provides demographic information about[64].

We consider a Bing user "active" in a given month if they issue at least 30 queries in that month. For most (over 90%) of queries, Bing estimates the ZIP code, county, and state from which the query originates. Based on an active user's query-level ZIP codes from the month, we assign the user to their mode ZIP code if the mode accounts for at least 10 and at least 25% of these queries (with the same rules for assigning county and state). We assume the mode is the user's likeliest home location from this month and include these additional requirements to avoid assigning users to locations that they just happened to visit and query from, but do not live in. Focusing on active users with a larger number of queries also improves our ability to reliably assign a user to a location. We estimate $N(b, z)$, the number of active Bing users from ZCTA $z$, as the average number of active users assigned to $z$ over the months in our study period (February to August 2021). In most instances, there is a one-to-one mapping from ZCTA to ZIP code, but for the ZCTAs that contain multiple ZIP codes, we set $N(b, z)$ to the sum of average user counts over those ZIP codes. We also acquire $N(z)$, the population size of $z$, from the 2020 5 year American Community Survey[65]. Finally, we estimate the ZCTA's coverage $\Pr(b=1|z)$ as $\frac{N(b,z)}{N(z)}$.

**Computing vaccine intent rates with inverse coverage.** Recall that our goal is to estimate $\tilde{p}(v, z) = \frac{\Pr(\hat{v}=1|z)}{\Pr(b=1|z)}$. To estimate $\Pr(\hat{v}=1|z)$, we apply our vaccine intent classifier to all queries and clicks of active Bing users. This produces $N(\hat{v}, z)$, the number of active Bing users from $z$ for

whom we detect vaccine intent. Then,

$$\tilde{p}(v,z) = \frac{\Pr(\hat{v}=1|z)}{\Pr(b=1|z)} = \frac{\frac{N(\hat{v},z)}{N(z)}}{\frac{N(b,z)}{N(z)}} = \frac{N(\hat{v},z)}{N(b,z)}. \qquad (4)$$

This is an intuitive result: our estimate for the vaccine intent rate in $z$ is the number of active Bing users with predicted vaccine intent divided by the total number of active Bing users. We can use these ZCTA-level rates to characterize demographic trends, such as by computing correlations between $\tilde{p}(v,z)$ and some demographic of $z$, such as its percentage aged 65 and over. It is also often useful to aggregate $\tilde{p}(v,z)$ over sets of ZCTAs, e.g., to the state or county-level. To compute the vaccine intent rate for a set $Z$ of ZCTAs, we simply take the population-weighted average:

$$\tilde{p}(v,Z) = \frac{\sum_{z \in Z} N(z) * \tilde{p}(v,z)}{\sum_{z \in Z} N(z)}. \qquad (5)$$

For example, we use estimated state and county vaccine intent rates to compare against reported vaccination rates from the CDC ("Comparison to reported vaccination rates"). This population-weighted average is equivalent to post-stratification[82], a common technique for adjusting non-representative survey responses to match known population totals, where we treat each ZCTA as a post-stratum.

**Bias in vaccine intent classifier.** Our primary source of bias is uneven Bing coverage, which we found can vary by > 2x across ZCTAs. However, after correcting for Bing coverage, we also want to know that our classifier does not significantly contribute to additional bias. To do this, we must establish that our classifier's true and false positive rates do not vary significantly or systematically across regions. The challenge is that we cannot perfectly evaluate our classifier's true or false positive rates, because we do not know all true positives or true negatives. However, we can approximate these metrics based on the labeled URLs that we do have and furthermore make methodological decisions that encourage similar performance across groups.

**Step 1: Personalized PageRank for URL candidates.** Recall that in the first step of our pipeline, we generate URL candidates for annotation by propagating labels from vaccine intent queries to unlabeled URLs in query-click graphs ("Personalized PageRank for URL candidates"). Since all URL candidates then go through manual inspection in Step 2, we do not have to worry about the false positive rate at this stage. However, we do need to worry about the true positive rate (i.e., recall). For example, if we only kept COVID-19 vaccine registration pages for pharmacies that are predominantly in certain regions, then we could be significantly likelier to detect true vaccine intent for certain states over others. So, through the design and evaluation of our label propagation techniques, we aim to ensure representativeness in vaccine intent across the US.

The most important design decision is that we construct query-click graphs *per state*, then we run S-PPR per graph and take the union over states of top URLs as our set of URL candidates. Running this process separately for each state allows us to capture how vaccine intent varies regionally, with state-specific programs and websites for scheduling the vaccine (Table S1). To demonstrate the risks of not using a state-specific approach, we try an alternative approach where we construct a joint graph that combines the queries and clicks for 6 states, chosen to vary in graph size and US region (the same 6 states as those used in the pre-training experiments of Table S2). To represent our union approach, we take the union over these 6 states of the top 200 URLs per state, which results in 935 URLs. We compare this to a joint approach, where we take the top 935 URLs from running S-PPR on the joint graph. To evaluate each approach, we compute the

proportion of each state's top $N$ URLs that are kept across different values of $N$. While we cannot be sure that every URL in the state's top $N$ is truly vaccine intent, from our annotation results, we saw high positive rates for top-ranking URLs (Fig. S2), so we would like to see similar recall at these ranks.

By design, our union-over-states approach ensures equivalent, 100% recall up to $N = 200$ for all states (Fig. S5, left). In comparison, we find that the joint approach yields different recalls as early as $N = 30$, with much higher recall for large states than small states (Fig. S5, right). For example, it keeps < 80% of Wyoming's URLs around rank 50 and <60% around rank 100, while keeping 100% of Tennessee's throughout. Furthermore, even past $N = 200$, where our union-over-states approach no longer has guarantees, we find that it still achieves far more similar recalls between states than the joint approach. Thus, our design decisions enable similar recalls between states, which helps to reduce downstream model bias. We also cast a wide net when constructing query-click graphs (taking all queries and clicks that co-occur in a session with any query that includes a COVID-19 or vaccine-related word), which may also improve recall and reduce bias, in case our choice of initial keywords was not representative of all vaccine intent searches across the US.

**Step 3: expanding vaccine intent URLs with GNNs.** In the third step of our pipeline, we use GNNs to expand our set of vaccine intent URLs beyond the manually labeled ones. We would like to see that the performance of GNNs is strong across states, to ensure that the GNN is not creating additional bias when expanding the URL set. We showed in "Graph neural networks for URL expansion" that, after incorporating pre-training on S-PPR rankings for smaller states, GNNs could achieve AUCs above 0.90 for all 50 states (Fig. S4, left). The main metrics of interest when considering bias, however, are the true and false positive rates (TPRs and FPRs). Unlike AUC, which is evaluated across decision thresholds, TPR and FPR depend on the chosen threshold $t$ above which data points are predicted to be positive. In our setting, we set $t = \max(t_{med}, t_{prec})$, since we required new vaccine intent URLs to score above these two thresholds (in at least 6 out of 10 trials): (1) $t_{med}$, the median score of positive URLs in the test set and (2) $t_{prec}$, the minimum threshold required to achieve precision of at least 0.9 on the test set. Then, we estimate TPR as the proportion of positive URLs in the test set that score above $t$ and FPR as the proportion of negative URLs in the test set that score above $t$.

We find that TPR is highly similar across states and hovers around 0.5 for all states (Fig. S4, middle). This is because in almost all cases, $t_{med}$ is the higher of the two thresholds and thus the value of $t$, so the true positive rate lands around 0.5 since $t_{med}$ is the median score of the true positives. FPR is also highly similar across states and very low (around 0.01; Fig. S4, right), which suggests that the quantity we estimate, $\tilde{p}(v,r)$, is not only a linear function of the true vaccine intent rate, $p(v,r)$, but also approximately proportional to it (Eq. (3)). The low FPR is encouraged but not guaranteed by our second threshold, $t_{prec}$. This threshold ensures that precision is over 0.9, which is equivalent to the false positive rate *among the predicted positives* being below 0.1, which typically corresponds to low false positive rates over all true negatives (which is what FPR measures). The GNN's similar AUCs, TPRs, and FPRs across states, as well as the equivalent recalls in our label propagation stage, increase confidence that our classifier is not adding significant bias to our estimates. In this section, we focused on states, since it was natural to evaluate performance per state due to the state-specific query-click graphs and models, and since we expect the expression of vaccine intent to vary most systematically per state due to state-specific vaccine programs and policies. In the following section, we continue this analysis by comparing our final $\tilde{p}(v,r)$ estimates per state to CDC vaccination rates, but we also test out finer-grained evaluations, including vaccination rates over time and rates at the county-level.

**Comparison to reported vaccination rates**

**Vaccination rates across states.** The CDC releases vaccination rates at the state and county levels. First, we compare against cumulative state-level vaccination rates. As our measure, we have $\tilde{p}(v, s)$ per state $s$, the cumulative vaccine intent rate in the state up to August 31, 2021, computed as described in Eq. (5). On the CDC side, we use the cumulative proportion of population fully vaccinated by August 31, 2021. We use percent fully vaccinated, which means completing the second dose of a 2-dose series or completing the first dose of a single-dose series, instead of percent with at least one dose, since the CDC reported the latter can be overestimated[83]. We find a strong Pearson correlation between these cumulative rates, with $r = 0.86$ (Fig. 1c). Notably, we find that the correlation drops to $r = 0.79$ if we do not correct for Bing coverage in our estimates and use a naive estimate instead that divides the total number of active users with vaccine intent by the total population size of the state (both summed over ZCTAs in the state), $\frac{\sum_z N(\hat{v},z)}{\sum_z N(z)}$. We also find that each step of our classification pipeline improves the correlation with CDC (Table 1): if we only use the seed set queries identified by our regex, $r = 0.62$; if we use the queries plus the URLs identified from manual annotation, $r = 0.80$. Furthermore, each step of our pipeline substantially increases the number of vaccine intent users we detect, which provides additional power for our downstream analyses.

**Vaccination rates over time.** We also compare against vaccination rates over time. For a given state $s$ and day $t$, we can compute $\tilde{p}(v, s, t)$, the vaccine intent rate on day $t$; this is equivalent to Eq. (5), except we only count users who showed their *first* vaccine intent on day $t$ instead of the cumulative count of users who have shown vaccine intent up to day $t$. We compare our metric to the daily proportion of individuals getting their first dose, since we expect the timing of the first vaccine intent to align more with the first dose than other doses, with a possible lag time. We can calculate the daily proportion for first dose from the cumulative at-least-one-dose time series provided by the CDC (by subtracting the cumulative count for day $t$ from day $t-1$). Furthermore, we apply 1 week smoothing to both daily time series (taking the average from day $t-6$ to $t$, inclusive), to smooth out daily effects such as potential underreporting on the weekends. For this analysis, we compare time series from April 1 to August 31, 2021, leaving out the months of February and March since we expect that vaccine intent was less correlated with actual vaccination rates early in the vaccine roll-out, since individuals would seek the vaccine but not be able to receive it yet (e.g., because they were not eligible or because there was not enough supply). Then, we compute the Pearson correlation between the smoothed daily time series, allowing the CDC time series to lag behind the vaccine intent time series by $l = \{0, 1, \cdots, 21\}$ days.

For each state, we compute the maximum correlation possible using the optimal lag. We observe strong temporal correlations, with correlations above 0.7 for 48 states and a median correlation of $r = 0.89$. We also observe a substantial lag between the vaccine intent and CDC time series, with a median optimal lag of 10 days (50% CI, 7.5-14.5); without any lag, the median correlation drops to $r = 0.78$. In Fig. S6, we visualize the vaccine intent and CDC vaccination time series for the 15 largest states in the US. Almost all of the correlations are strong except for North Carolina, where the CDC time series shows an anomalous peak at the beginning of July 2021. This peak seems to be an artifact of the CDC data, since it is so much larger and sharper than any other peak we see and cannot be easily explained by concurrent news or events. These anomalies further motivate the need for complementary data sources, such as our classifier, that can also track vaccine seeking over time. We also see from this figure that the optimal lag varies across these states, but the CDC time series is always at least 1 week behind vaccine intent.

We also test whether daily vaccine intent Granger-causes daily CDC vaccinations. Granger causality tests whether using past values of vaccine intent and past values of CDC vaccinations to predict current CDC vaccinations significantly outperforms only using past values of CDC vaccinations. Using a lag of 1 week, we find that vaccine intent *does* Granger-cause CDC vaccinations in the majority of states ($p < 0.05$), using three different statistical tests (Table S4). Results are similar for lags of 2, 3, and 4 weeks. These results demonstrate the predictive utility of our vaccine intent signals, beyond correlations, which can help policymakers preempt vaccinations and meet population needs more efficiently. In the SI, we discuss how agreement between search signals and real-world trends can change over time, as they did for Google Flu Trends[37], and how we mitigate such temporal drift.

**Vaccination rates across counties.** We also compare to CDC county-level vaccination rates. CDC county-level data are imperfect: for example, we find that there are 53 counties with no data and 276 counties reporting < 1% of the population fully vaccinated, which is unrealistic given that over 60% of the US population was vaccinated by this time. We also find 628 counties where the "completeness percent", i.e., the percent of vaccination records that include county of residence, is < 80%. Following prior work using these data[12], we exclude these incomplete counties from our analysis. In total, out of 676 counties with missing or unreliable CDC data, we are able to provide vaccine intent estimates for 590 of them, demonstrating our classifier's ability to fill in gaps in CDC reporting. On the remaining counties, we compare our estimated vaccine rates $\tilde{p}(v, c)$ per county $c$ to the CDC's fully vaccinated rates, cumulative up to August 31, 2021.

The Pearson correlation, weighted by square root of county population, is $r = 0.68$ (Fig. S7), which is lower than the state-level correlation but still largely in agreement. Notably, we achieve higher correlations on counties where we expect higher-quality CDC reporting, which are counties with higher completeness percentages and larger populations, where reported proportions are less noisy. If we remove the constraint on completeness percent, our correlation drops to $r = 0.54$. If we keep the completeness constraint at 80% but remove weighting by population size, our correlation drops to $r = 0.58$. These observations suggest that discrepancies between our estimates and CDC data are at least in part driven by issues in CDC reporting, since our agreement improves on counties with higher-quality reporting.

**Vaccination rates across ZIP codes.** The CDC does not report ZIP-level vaccination rates, which is one motivation for developing our vaccine classifier, since it can estimate finer-grained rates of vaccine intent from search logs. To evaluate our classifier at finer granularities, we need to rely on local reporting of ZIP-level vaccination rates, which only occurred in a handful of US states and cities. We compare to two sources of ZIP-level vaccination rates. First, we acquire historical ZCTA-level vaccination rates in California, reported by the California Department of Public Health[70]. We compare our estimated vaccine rates $\tilde{p}(v, z)$ per ZCTA $z$ to California's reported rates of percent fully vaccinated, cumulative through August 31, 2021. The California data reports 1,764 ZCTAs, although 162 are missing values for percent fully vaccinated, 2 are unrealistically low (<1%), and 15 are unrealistically high (≥100%). Of the remaining ZCTAs, we have vaccine intent estimates for 1308 of them: over these ZCTAs, the Pearson correlation, weighted by square root of ZCTA population, is $r = 0.55$ (Fig. S8a). We also compare to ZCTA-level vaccination rates from 16 large US cities, cumulative through September 2021, which were compiled by the Big Cities Health Coalition's COVID-19 Health Inequities in Cities Dashboard[71] and made available by Bilal et al.[72]. As stated in Bilal et al.[72], they "calculated the proportion of fully vaccinated adults in 866 zip code tabulation areas (ZCTAs) of 16 large US cities: Long Beach, Los Angeles, Oakland, San Diego, San Francisco, and San Jose, all in

California; Chicago, Illinois; Indianapolis, Indiana; Minneapolis, Minnesota; New York, New York; Philadelphia, Pennsylvania; and Austin, Dallas, Fort Worth, Houston, and San Antonio, all in Texas." Among these ZCTAs, 25 have unrealistically high rates (≥100%). Of the remaining ZCTAs, we have vaccine intent estimates for 837 of them: over these ZCTAs, the Pearson correlation, weighted by square root of ZCTA population, is $r = 0.47$ (Fig. S8b).

While these ZCTA-level correlations are not as high as the state or county-level correlations, they are still substantial, and similar in magnitude to prior work that presented search data as a measurement or forecasting tool. For example, Chancellor and Counts[84] use search data to measure employment demand and show a correlation of $r = 0.47$ between job searches and county-level employment rates, and Lin et al.[85] use search data to forecast US domestic migration, and report correlations of $r = 0.72$, 0.44, and 0.39 between migration intent and state-level inflow, outflow, and net migration respectively. Furthermore, the lower correlations observed at the ZCTA-level are due to increased noise in not only our vaccine intent estimates, but also in vaccine reporting. Part of the noise is due to trying to estimate proportions from smaller population sizes, since ZCTAs are around $10 \times$ smaller than counties. As in our county-level analysis, we find that if we do not weight by population size, the correlations drop by 3-4 points for both datasets, which supports the theory that our correlations may be lower due to noise in smaller population sizes. There are also other sources of noise and bias that are unique to the ZCTA-level reported vaccination rates. For example, the unrealistically low and high rates that appeared in both ZCTA-level datasets do not appear in state-level or county-level rates (with completeness above 80%). Bilal et al. also note that "the data for Philadelphia, Chicago, and New York City do not, to our knowledge, include residents who were vaccinated outside of their respective cities", which results in not only underestimating rates of vaccination in those cities but also biased reporting, due to certain populations (e.g., higher income) being likelier to be vaccinated outside of their cities[72]. The California Department of Public Health also notes limitations in their data: for example, reported vaccination coverage may exceed 100% for some ZCTAs, which "may be a result of many people from outside the county coming to that ZCTA to get their vaccine and providers reporting the county of administration as the county of residence, and/or the DOF estimates of the population in that ZCTA are too low." In contrast, our vaccine intent estimates are not affected by whether an individual travels outside of their home ZCTA to receive their vaccine, since we assign home ZCTA based on the user's mode ZIP code from over 30+ queries in that month. Furthermore, our estimates do not show the same unrealistic values: without any smoothing or clipping, none of our estimated rates are below 1% or above 100% for the ZCTAs in either dataset, due to our careful inclusion criteria for active Bing users and vaccine intent.

In summary, our ZCTA-level vaccine rates offer several advantages over traditional data sources. First, we provide a *unified* framework for estimating ZCTA-level rates, over all US states, instead of relying on a patchwork of ZCTA-level rates from some states and cities, each with their own reporting systems and biases. Second, since only a handful of states and cities currently report rates, our data provides, to the best of our knowledge, the *largest* ZCTA/ZIP-level dataset of COVID-19 vaccine intent or vaccination rates in the US. Currently, nationwide vaccination rates are only available at the county-level, which are insufficient for capturing substantial heterogeneity within counties (Fig. 2b, Fig. S11). Third, we provide time-varying vaccine rate estimates, while Bilal et al. note that some cities do not provide longitudinal ZCTA-level vaccination rates[72], which misses important changes over time in vaccination rates, as eligibility and policies (e.g., vaccine mandates) evolved. Thus, while our finer-grained estimates are by no means perfect, they provide many benefits that make them a useful, complementary data source to traditional reporting from the CDC or local public health departments, which is slower, coarser-grained, and has limitations of its own.

## Comparison to Google search trends

**Search trends over time.** Following prior work using Bing data[23], we compare Bing and Google queries to evaluate the representativeness of Bing search data. First, we compare daily search interest in the US over our studied time period from February 1–August 31, 2021. Google Trends provides normalized search interest over time on Google, such that 100 represents the peak popularity for that time period, 50 means the term is half as popular, and 0 means "there was not enough data for this term." To match this, for a given query, we compute the total number of times it was searched on Bing in the US per day, then we divide by the maximum number and multiply by 100. Again, we apply 1 week smoothing to both the Bing and Google time series. We do not correct the Bing time series with Bing coverage here, since we cannot correct the Google time series with Google coverage, and we want the time series to be constructed as similarly as possible.

We choose 30 representative vaccine intent queries from the top 100 vaccine intent queries, where we choose one standard query for each pharmacy that appears (e.g., [cvs covid vaccine]) and one for each location-seeking query (e.g., [covid vaccine near me]), and drop variants such as [cvs covid vaccines] and [covid 19 vaccine near me]. Over these queries, we observe strong Pearson correlations, with a median correlation of $r = 0.95$ (90% CI, 0.88-0.99) (Fig. S9). These correlations are similar to those reported by Suh et al.[23], who conduct an analogous longitudinal analysis comparing Bing and Google search trends on COVID-related queries and report correlations from $r = 0.86$ to 0.98. Remaining discrepancies between Bing and Google are likely due to differences in the populations using these search engines, as well as potential unreported details on how Google normalizes their search interest trends (e.g., Google may be normalizing differently for [covid vaccine near me], which shows unusual peaks in Google trends and is the the only query for which we do not observe a strong correlation).

**Search trends across states.** Google also provides normalized search interest across US states, where search interest is defined as the fraction of searches from that state that match the query and search interest is normalized across regions such that 100 represents maximum popularity. To imitate this process, we first assign each vaccine intent query to a state based on where the query originated. Then, we approximate the total number of queries (all queries, not just vaccine intent) from each state by summing over the query counts of the active users assigned to each state. We compute the fraction of queries from each state that match the query, then we divide by the maximum fraction and multiply by 100 to normalize across states.

We observe strong Pearson correlations in this analysis too, with a median correlation of $r = 0.95$ (90% CI, 0.57-0.99) across the same 30 vaccine intent queries (Fig. S10). The correlations tend to be stronger on the pharmacy-specific queries, where certain regions dominate, compared to general location-seeking queries such as [covid vaccine near me], which are trickier since they follow less obvious geographical patterns. For the pharmacy-specific queries, we also observe substantial heterogeneity in terms of which region dominates. For example, [publix covid vaccine] is more popular in southern states, with Florida exhibiting the maximum normalized search interest on Google (100), followed by Georgia (26) and South Carolina (20). Meanwhile, [cvs covid vaccine] is more popular in the Northeast, with the top states being Massachusetts (100), New Jersey (96), Rhode Island (90), and Connecticut (65). These differences, reflected in the Bing search trends too, once again highlight the need for regional awareness and representativeness when developing our vaccine intent classifier.

The strong correlation between Bing queries and Google queries is reassuring, demonstrating that trends on Bing are not abnormal compared to Google, which is the more popular search engine.

However, note that it would not be possible to simply use Google Trends to recreate our vaccine intent classifier. First, our classifier uses queries and clicks and finds that adding clicks substantially improves the classifier's performance (Table 1), while Google Trends only provides query information. Furthermore, it would be hard to estimate the proportion of the region's population showing vaccine intent from Google Trends, since they provide normalized query interest as a fraction of all queries searched, as opposed to our much more exact measurement of what proportion of active Bing users have shown vaccine intent. Finally, Google Trends do not provide county- or ZIP-level data, and one of our classifier's key strengths is its ability to make finer-grained estimates, compared to CDC reporting. So, it would not be possible to use Google Trends to recreate our vaccine intent classifier, but the correlation between Bing and Google is reassuring.

### Taxonomy of vaccine concerns on search

In this section, we describe how we match vaccine holdouts and vaccine early adopters ("Identifying matched holdouts and early adopters"), then how we construct a hierarchical taxonomy of vaccine concerns, based on the their clicks from April to August 2021 ("Constructing a taxonomy of search concerns").

**Identifying matched holdouts and early adopters.** Our study period covers search logs from February 1–August 31, 2021. First, we apply our vaccine intent classifier to the entire study period and identify 7.45M users who have expressed vaccine intent through queries and/or clicks. Among those users, we define early adopters as those who showed their first vaccine intent before May (i.e., between February 1 and April 30, 2021) and vaccine holdouts as those who waited until July to show their first vaccine intent (i.e., between July 1 and August 31, 2021). We did not consider as holdouts those who never showed vaccine intent during our study period, since those users may have gotten their vaccine in ways that are not visible via search data, e.g., a walk-in appointment. In comparison, individuals who did not show their first vaccine intent until July 2021 likely did not receive the vaccine before. We choose July as a cutoff since all US residents aged 16 and older were eligible for the vaccine by April 19[54], so those who waited until July to seek the vaccine were holding out. Furthermore, to improve our ability to detect true holdouts, we require holdouts and early adopters to be active (i.e., issued at least 30 queries) in every month during the study period, since if users were not active on Bing before July or August, their apparent lack of vaccine intent could be explained simply by low Bing usage during the earlier months.

To reduce potential confounding, we match each vaccine holdout to a unique vaccine early adopter from the same county, since we know that the populations seeking vaccination changed over time (Figs. 1d and S18) and we do not want our comparisons to be overpowered by regional or demographic differences. We also match on Bing usage, by requiring that the holdout's and early adopter's average monthly query counts during the study period do not differ by >10 queries. We match on query count since our estimated time of first vaccine intent may be more delayed for individuals with less frequent Bing usage, so being labeled a holdout may be correlated with using Bing less frequently, which may reflect latent variables that also affect individuals' search behaviors. To implement matching, we construct a bipartite graph between holdouts and early adopters, where an edge between a holdout and early adopter exists if that early adopter is a valid match for the holdout (they are from the same county and their average query counts are within 10 of each other). Then, we run the Hopcroft-Karp algorithm on this graph, which finds the maximum matching, i.e., the largest set of edges where no two edges share an endpoint[86]. Since the number of early adopters greatly outnumbers the number of holdouts, we are able to match 98% of our holdouts using this approach, resulting in 212,283 matched pairs.

**Constructing a taxonomy of search concerns.** To analyze the vaccine concerns of holdouts and early adopters, our goal is to organize the vaccine-related URLs that they click on into a rich taxonomy. We focus on their clicks from April to August 2021, since from April to June we can compare their vaccine concerns during the period while holdouts were eligible for the vaccine but still holding out, and from July to August, we can study how holdouts' concerns change as they finally express vaccine intent for the first time. To construct our taxonomy, we combine computational and manual approaches: first, we use machine learning techniques to partition the URLs into clusters, and then we manually label each cluster and develop our three-tiered taxonomy of categories, subcategories, and URL clusters.

**Automatically partitioning URLs into clusters.** We begin by gathering all vaccine-related URLs (i.e., containing "vaccin" or "vax") that were clicked on by holdouts or early adopters from April to August 2021. We drop all of the vaccine intent URLs, including the manual ones labeled by AMT or by regular expressions ("Annotation on Amazon Mechanical Turk") and the ones discovered by GNNs ("Graph neural networks for URL expansion"). We also drop internal Microsoft links (e.g., ads) and URLs that were clicked on by < 5 users (summed over holdouts and early adopters). After filtering, we are left with 32,811 vaccine-related URLs. We construct the query-click graph for these URLs, based on the queries and clicks of holdouts and early adopters from April to August 2021. Since we only want to partition URLs, we then collapse the query-click graph to construct the URL co-click graph, where a weighted edge between two URLs indicates how connected these two URLs are by common queries. Then, we partition the URLs into clusters by running the Louvain community detection algorithm[50] on the co-click graph. Empirically, we find that clusters of 100–500 URLs are the most useful: they are large enough to represent substantial topics (as opposed to, for example, individual news stories), but not so large that their focus is unclear. We tune the Louvain resolution parameter $\gamma$, which controls how much the algorithm favors larger as opposed to smaller communities, to maximize the number of clusters within this range. We find that $\gamma = 17$ allows us to achieve the largest number of clusters of size 100–500.

**Manually constructing our taxonomy.** We begin by labeling the clusters with at least 100 URLs, of which there are 79 when $\gamma = 17$. Each author labels the clusters independently, viewing a uniform random sample of 30 URLs from each cluster. To aid our labeling process, we also view the 10 most frequent queries for each cluster, which we obtain by summing over all queries that led to clicks on URLs in the cluster. When labeling, each author writes a free-text description of each cluster and marks whether it is clear. We find that our automatic approach produces remarkably coherent topics and that the majority of clusters are clear. In Table S5, we provide a sample of clusters. From the top query and most frequently clicked URLs, we observe distinct topics covered in each cluster: one on CDC masking guidelines after vaccination, one on the Vaccine Adverse Event Reporting System (VAERS)[87], one about religious exemptions for COVID-19 vaccine requirements, and one about side effects of the Johnson & Johnson vaccine.

Based on our descriptions of the clusters, we identify 8 top categories and 36 subcategories of vaccine concerns (Fig. 3). For example, under Vaccine Safety, we include the subcategories of normal side effects (e.g., sore arms), severe side effects (e.g., blood clots), concerns about reproductive health, fear of vaccine-caused deaths, "eerie" fears (e.g., myths about vaccine shedding or becoming magnetic[53]), vaccine development (e.g., pace of development, ingredients in the vaccine), and FDA approval. As we show in the following section, these fine-grained subcategories allow us to study nuances in vaccine concerns; for example, holdouts and early adopters are both concerned about vaccine safety, but focus on different aspects of it. Finally, we take a

second pass through the URL clusters to sort them into subcategories, allowing each cluster to belong to at most 2 subcategories. For example, for the cluster about religious exemptions (Table S5), we sort it into both the religious concerns and exemptions subcategories. During this second pass, we label all clusters with at least 30 URLs. For the clusters that are unclear, we rerun Louvain community detection on the cluster (with the default $\gamma = 1$, since the number of URLs being partitioned is much smaller) to try to identify smaller groups of URLs that we can assign to subcategories. We are able to assign most clusters to subcategories, but there are some that we leave out, since they cover miscellaneous topics such as one-off news stories or specific interests (e.g., how to store vaccines) that do not clearly belong to any of our subcategories. At the end of our process, our constructed taxonomy consists of 24,726 URLs (75% of all 32,811 vaccine-related URLs).

## Main analyses

**Granular trends in vaccine seeking.** In Fig. 2, we visualize our estimates of ZCTA vaccine intent rates. For privacy reasons, we focus our analyses on ZCTAs where the number of active Bing users $N(b, z) \geq 50$ and the Census population size $N(z) \geq 50$, keeping 20,899 ZCTAs and covering 97% of the US population. We estimate coverage-corrected vaccine intent rates $\tilde{p}(v, z)$ per ZCTA $z$ following Eq. (4).

**Maps and within-county heterogeneity.** For our map visualizations, we use ZCTA, county, and state shapefiles from the 2020 US Census[67]. In Fig. 2a, we visualize vaccine intent rates for the entire US. Since we cannot estimate vaccine intent rates for all ZCTAs, first we visualize vaccine intent rates per county (following Eq. (5)), then we overlay ZCTA vaccine intent rates. Additionally, we overlay the boundaries of the states in white, to emphasize state-level trends as well as reveal substantial heterogeneity in vaccine intent rates within the same state. To explore this heterogeneity in greater detail, in Fig. 2b, we zoom in on the five counties in New York City, corresponding to the five boroughs: Manhattan (New York County), Queens (Queens County), Brooklyn (Kings County), Bronx (Bronx County), and Staten Island (Richmond County). ZCTAs are well-represented here, given the population of the city, so we only visualize ZCTA vaccine intent rates without county-level rates. Also, instead of drawing state boundaries in white, we draw county boundaries in white, to emphasize trends per county and heterogeneity within each county. For example, we see that Manhattan and Queens have higher estimated rates of vaccine intent, and within Queens, ZCTAs in the northern half have higher rates, aligning with reported local vaccination rates in New York City earlier in the pandemic[46].

In fact, over all ZCTAs and counties, we find large heterogeneity *within* counties that often exceeds heterogeneity *between* counties. Across counties, the standard deviation in vaccine intent rates is 0.067 and the coefficient of variation is 0.285. In Fig. S11, we visualize the standard deviations and coefficients of variation *within* counties (over ZCTAs), for all counties where we have estimated vaccine rates for at least 5 ZCTAs, and we see that for many of these counties, their within-county variation exceeds the between-county variation. These levels of variation within counties are supported by the real-world ZCTA-level datasets that we have; for example, in the California Department of Public Health dataset[70] of ZCTA-level vaccination rates (cumulative up to August 31, 2021), the coefficients of variation within counties ranges from 0.095 to 0.647, similar to the range that we find in our estimated vaccine intent rates (Fig. S11, right).

**Measuring demographic trends.** To characterize this heterogeneity, we compare ZCTA vaccine intent rates to demographic variables. First, we measure the Pearson correlation between vaccine intent rate and each demographic variable, with correlation weighted by the square root of the ZCTA population. In Fig. 2c, we plot the results, ordered by strongest positive to strongest negative correlation, with 95%

confidence intervals[88]. Our results agree with prior literature, finding positive correlations with percent with Bachelor degree, median income, population per square meter, percent 65 and over, percent Asian, percent White, and percent female, and negative correlations with percent Republican, percent under 18, percent Black, and percent Hispanic[16,47–49]. In the SI, we also conduct supplementary analyses of demographic trends, by separating ZCTAs into their bottom and top quartile by demographic variable (e.g., median income). Then, we can compare the average vaccine intent of ZCTAs in the bottom versus top quartile, cumulatively (which yields similar results to correlations) and over time.

To investigate geographic differences in demographic trends, we also measure correlations per state (only including the ZCTAs in the state) for the 10 largest states in the US. For this finer-grained analysis, we drop percent Republican, since we only have vote share at the county-level, but we keep all other demographic variables, which we have per ZCTA. We find that correlations are mostly consistent in sign across states, but the magnitude differs significantly (Fig. S12). For example, the positive correlation with percent 65 and over is around 2x as high in Florida as it is in the second highest states, reflecting the large senior population in Florida and the push for seniors to get vaccinated. In most states, we also see positive correlations for percent Asian and percent White, and negative correlations for percent Black and percent Hispanic, aligning with prior research on racial and ethnic disparities in COVID-19 vaccination rates[89,90]. Positive and negative correlations for race are particularly strong in certain states, including New York and Florida for percent White/Black, and California and New York for percent Hispanic.

**Analyses of news consumption and vaccine concerns.** In this section, we compare the news consumption and vaccine concerns of holdouts and their matched early adopters from April to June 2021. We focus on this time period since we want to compare search behavior from when holdouts were eligible for the vaccine but still had not shown vaccine intent (which they showed in July and August 2021). In this section, we describe the following studies: (1) comparing the news consumption of holdouts versus matched early adopters, using labels of news trustworthiness from Newsguard, (2) comparing the vaccine concerns of holdouts versus matched early adopters, using our new taxonomy of vaccine concerns ("Taxonomy of vaccine concerns on search"), (3) analyzing variation in vaccine concerns among holdouts, when grouped by demographics, (4) discovering different holdout "profiles", or clusters, from their individual vaccine concerns.

**Click ratios comparing two groups.** In several analyses, we compare click probabilities for two groups, such as the probability that a holdout versus an early adopter clicks on untrusted news. For each group $g$, such as holdouts, first we gather all their relevant clicks $R_g$ (e.g., news-related, vaccine-related) from the time period of the analysis. Then, we identify the "positive" subset of clicks $S_g \subseteq R_g$, such as clicks on untrusted news or a specific vaccine subcategory. Then, we compute $p_g$, the weighted average probability over clicks that the click is in the positive set, weighted by the user's ZCTA's inverse Bing coverage ("Coverage-corrected vaccine intent rates"):

$$p_g = \frac{\sum_{x_i \in R_g} \mathbb{1}[x_i \in S_g] \frac{N(z_i)}{N(b,z_i)}}{\sum_{x_j \in R_g} \frac{N(z_j)}{N(b,z_j)}}, \tag{6}$$

where $z_i$ represents the ZCTA of click $x_i$'s user. To compare two groups, such as holdouts and early adopters, we take the ratios of their click probabilities. We compute bootstrapped CIs for these ratios by repeatedly resampling the clicks in each group (with replacement), recomputing the group averages, and recomputing the ratio between group averages. Then, from the resulting distribution of ratios,

computed over 1000 bootstrapped samples, we report the 2.5th and 97.5th percentiles as the 95% CI.

We compare to an alternative measure that computes the fraction of clicks in the positive set *per user*, then computes the weighted average over users, weighting again by their ZCTA's inverse Bing coverage. We find in practice that the measures result in very similar trends ($r > 0.95$), which is expected, since we have over 200,000 users per group and most users have clicks (since we only keep users who are active in every month during our study period), so our measure is not dominated by a few users' clicks even if we do not compute user-specific fractions first. So, we use our measure $p_g$ for simplicity, which is more straightforward to compute and bootstrap.

**News consumption of holdouts versus early adopters.** In this analysis, we compare the news consumption of holdouts versus matched early adopters from April to June 2021. We use labels from Newsguard, which assigns numerical trust scores to news domains indicating their trustworthiness, along with categorical rankings, where domains with scores above 60 are considered trusted and below 60 are considered untrusted ("Datasets"). We have Newsguard labels for 7226 news domains, out of which 2744 are labeled as untrusted. For each group $g$, the relevant set of clicks $R_g$ includes all clicks on news domains from April 1 to June 30, 2021. To prevent leakage, we remove the clicks on vaccine intent URLs; this is so that we do not end up predicting greater rates of trusted news for early adopters simply because, by construction, they were clicking on vaccine intent URLs (which tend to be trusted, if they are news domains) during that time period while holdouts were not.

First, we compare the probabilities of clicking on untrusted news, so we set positive clicks $S_g$ to clicks on untrusted news domains, and compute average probabilities for holdouts and early adopters according to Eq. (6). We find that holdouts are 66.6% (95% CI, 65.6% -67.6%) likelier to click on untrusted news, compared to their matched early adopters. To compute click ratios for a specific news domain, we set $S_g$ to clicks on that domain, and compute weighted probabilities and ratios accordingly. We compute and visualize ratios for domains that receive at least 0.0001% of news clicks for both holdouts and matched early adopters, which leaves 3285 domains in Fig. 4a. We find that, over these 3285 domains, there is a substantial negative correlation between the domain's trust score from Newsguard and holdout ratio. The Pearson correlation, weighted by average proportion of clicks, is $r = -0.411$, meaning that as the trust score from Newsguard degrades, the likelier it was that holdouts clicked on the site, relative to matched early adopters. Furthermore, this effect becomes stronger among mainstream news: if we only include the top 100 news domains that received the largest average proportions of news clicks in our data, the correlation becomes $r = -0.565$. We describe supplemental analyses in the SI, including ratios for the most-visited news sites in the US (Table S8) and analyses that show that the overall trends are not only driven by a few outlier holdouts.

**Vaccine concerns of holdouts versus early adopters.** In this analysis, we compare the vaccine concerns of holdouts versus matched early adopters from April to June 2021. Our relevant set of clicks $R_g$ now includes each group's vaccine-related clicks (i.e., containing "vaccin" or "vax") during this time period. As in our news consumption analyses, we remove the clicks on vaccine intent URLs, since we used those URLs to label users as holdouts and early adopters, so we do not want to use the same URLs to characterize and compare their interests. To analyze click patterns for a given category of concerns (or subcategory), we set positive clicks $S_g$ to clicks on URLs belonging to that category, according to our taxonomy, and compute probabilities and ratios accordingly (Eq. (6)). In Fig. 4b, we visualize holdouts' click proportions over time for the 6 main categories: Vaccine Requirements, Community, Effectiveness, Safety, Incentives, and Information. We

report click ratios and 95% CIs for all categories and subcategories in Table S6, and visualize subcategory click ratios in Fig. 4c. In both Fig. 4b and c, categories/subcategories are ordered top to bottom and colored from yellow to dark purple in terms of most holdout-leaning to most early adopter-leaning.

**Variation in vaccine concerns among holdouts, by demographic.** Since we have individual clicks from holdouts, we can not only compare vaccine concerns across holdouts and early adopters, but also investigate variability in concerns *among* holdouts. First, we compare holdout concerns across demographic groups. For a given demographic variable, such as median income, we compute its median value across *all* ZCTAs in the US, split holdouts into those from ZCTAs above the median versus those from ZCTAs below the median, then compare the vaccine concerns of those two groups of holdouts (by measuring their click ratios, following Eq. (6)). We explore three key variables—proportion white, median income, and proportion Republican (based on the 2020 presidential election)—and we find significant variability across all three demographics.

When we split holdouts by proportion Republican (Fig. S13), we find that holdouts from more Democrat-leaning ZCTAs were far more interested in requirements around employee mandates and vaccine proof, which may be because jurisdictions run by Democrats were likelier to have vaccine requirements[51,52], while several Republican governors in fact banned such requirements. Meanwhile, holdouts from more Republican-leaning ZCTAs were more interested in eerie vaccine fears, vaccine-caused deaths, and vaccine incentives. When we split holdouts by median income (Fig. S14), we find that holdouts from higher-income ZCTAs were significantly more interested in vaccine requirements, vaccine rates, and anti-vaccine messages from experts and high-profile figures, while holdouts from lower-income ZCTAs were more interested in vaccine incentives and religious concerns about the vaccine. Finally, when we split holdouts by proportion white (Fig. S15), we find that holdouts from more white ZCTAs showed greater interest in vaccine incentives, vaccine-caused deaths, and comparing vaccines to natural immunity, while holdouts from less white ZCTAs were more interested in vaccine rates, proof of vaccine, and FDA approval. These significant differences reveal that vaccine concerns are not uniform across holdouts and vaccine hesitancy likely cannot be addressed through one-size-fits-all solutions.

**Discovering holdout "profiles" from individual vaccine concerns.** In addition to grouping holdouts by their ZCTA demographics, we can also directly analyze the individual concerns of vaccine holdouts. Here, we seek to understand, which types of concerns tend to co-occur for the same person, and can we discover different "profiles" by clustering holdouts based on their concerns? To do this, first we construct a matrix of holdout users by subcategories, where we record the number of clicks that the user made per subcategory from April to June 2021. For this analysis, we drop subcategories in the Availability category, which consists of seeking vaccine boosters, vaccines for children, and vaccine locations, since we want to focus on individuals making decisions about whether to receive the primary series COVID-19 vaccine for themselves. We keep all holdouts with at least 10 clicks in total on these subcategories, resulting in 546 holdouts, and normalize each row in our matrix to sum to 1. Thus, each holdout user $u$ is represented by a vector $\mathbf{v}_u = \{\pi_{u1}, \pi_{u2}, \cdots, \pi_{uS}\}$, where $\pi_{us}$ represents the user's relative likelihood of clicking on subcategory $s$ and $\sum_s \pi_{us} = 1$. Normalizing the rows enables us to compare holdouts based on their proportions of clicks, instead of overall number of clicks, but it transforms our data into compositional data, which introduces additional steps before clustering. Following standard practice for clustering compositional data[91], we apply an isometric ratio transform (ilr), since in its original compositional form, Euclidean distance (which is assumed by many clustering techniques, like k-means) is not an appropriate measure of

distance; then, we apply k-means clustering to the ilr coordinates. We test $k = 2$ to $k = 20$ and, using the heuristic "elbow method", choose $k = 4$ as a reasonable number of clusters. Then, for each of the four clusters, we compute the mean proportions over subcategories of the users *in* the cluster, divided by the mean proportions of users *not in* the cluster, and report those ratios to characterize what is unique in each cluster.

We find that four clear profiles arise (Fig. S16). In Cluster 1, with 149 users, we see increased interest in eerie fears (e.g., that microchips are inserted into the vaccine), anti-vaccine messages from scientific "experts", and skepticism about vaccine development and ingredients. All of these subcategories involve known misinformation and they represent what one might expect a stereotypical holdout to be concerned with. However, there are three more clusters of holdouts. In Cluster 2, with 115 users, we find increased interest in government policies around vaccines: vaccine incentives and vaccine requirements (travel restrictions, proof of vaccine, and employee mandates), along with seeking exemptions for these requirements. In Cluster 3, with 144 users, we see a much higher rate of interest in decision-making (e.g., articles reflecting the pros and cons of receiving the COVID-19 vaccine) and reading news about vaccine hesitancy; these holdouts appear to be grappling with the decision of whether to receive the vaccine or not. In Cluster 4, with 138 users, we see increased interest in side effects (normal and severe) and seeking information about the different vaccines (Moderna, Pfizer, and Johnson & Johnson). These users appear to be the closest to vaccine intent, since they are seeking information about specific vaccine brands and realistic side effects that occur after receiving the vaccine. These profiles illustrate four very different types of holdouts, who vary in their openness to the vaccine (e.g., Cluster 4 seems the most open) and their key concerns, which implies that policymakers need to go beyond one-size-fits-all solutions to address vaccine hesitancy. Instead, persuading different individuals will require different interventions: for example, discussing the vaccine's safety may help to address Cluster 1's concerns while Cluster 2 may be more convinced by vaccine requirements or incentives.

**Changes in vaccine concerns as holdouts approach vaccine intent.** Since we defined holdouts as those who waited several months to show vaccine intent but eventually did (in July and August 2021), we have the opportunity to study how holdouts' vaccine concerns *changed* leading up to their eventual vaccine intent. First, we analyze how holdouts' vaccine concerns change in the small window leading up to and following their expressed vaccine intent. We split holdouts' vaccine-related clicks from July to August 2021 into two groups: clicks when the holdout is within 3 days (either before or after) of expressing vaccine intent and clicks outside of that range. Since precision of vaccine intent timing is particularly important for this analysis, we focus on vaccine intent expressed either through a vaccine intent query or a manually labeled vaccine intent URL (either from AMT or regular expression). Then, as before, we compute ratios per subcategory, where the set of positive clicks $S_g$ are those that match the subcategory and we compute probabilities per group according to Eq. (6). We report click ratios and 95% CIs for all categories and subcategories in Table S7, and visualize subcategory click ratios in Fig. 4d. To facilitate comparison between Fig. 4c and d, we keep the ordering of subcategories the same. This design choice highlights how Fig. 4d nearly reverses Fig. 4c, meaning that near when holdouts express vaccine intent, their concerns become much more like the concerns of early adopters, with a few important differences (e.g., greater relative interest in Johnson & Johnson, less interest in vaccine rates).

We also conduct two supplementary analyses, which we describe in the SI: (1) a more detailed version of the first analysis, with ratios for each day relative to vaccine intent from -14 – +7 days; (2) a *predictive* study that tests whether changes in vaccine concerns and news consumption Granger-cause the timing of vaccine intent, i.e., provide

predictive power beyond using past values of vaccine intent. We find that changes in vaccine concerns and increase in trusted news consumption both Granger-cause the timing of vaccine intent. These findings may help to guide budgeted interventions from policymakers, as well as begin to explain *why* holdouts changed their minds, which may be studied in depth in the future with true causal analyses.

### Reporting summary
Further information on research design is available in the Nature Portfolio Reporting Summary linked to this article.

## Data availability
Our vaccine intent estimates and taxonomy of vaccine concerns are publicly available at https://github.com/microsoft/vaccine_search_study. Aside from Bing search logs, all of the data sources that we use are publicly available for download or purchase. Data from the US Census American Community Survey (https://www.census.gov/programs-surveys/acs/data.html), Census shapefiles for map visualizations (https://www.census.gov/cgi-bin/geo/shapefiles/index.php), state-level CDC vaccination rates (https://data.cdc.gov/Vaccinations/COVID-19-Vaccinations-in-the-United-States-Jurisdi/unsk-b7fc), county-level CDC vaccination rates (https://data.cdc.gov/Vaccinations/COVID-19-Vaccinations-in-the-United-States-County/8xkx-amqh), California ZCTA-level vaccination rates (https://data.chhs.ca.gov/dataset/covid-19-vaccine-progress-dashboard-data-by-zip-code), ZCTA-level vaccination rates from the Big Cities Health Coalition (https://github.com/usamabilal/COVID_Vaccines_Disparities/tree/main), and Google search trends (https://trends.google.com/trends/?geo=US) can be downloaded without cost. Data on US elections (https://uselectionatlas.org/BOTTOM/store_data.php) and Newsguard data (https://www.newsguardtech.com/solutions/newsguard/) can be purchased. The Bing search logs are not publicly available, for privacy and legal reasons. Subject to data protection guidelines, we plan to retain aggregated versions of the data (e.g., clicks on different vaccine concerns, aggregated over 200,000+ holdouts) indefinitely for scientific and academic purposes. To request access to the aggregated data, please contact the corresponding author, Dr. Eric Horvitz, who can be reached at horvitz@microsoft.com, with a clear description of how the data will be used and the purpose of the proposed study. The request will be reviewed and approved on a case-by-case basis by the Microsoft Research Release and Compliance team, at which point a license agreement will be drafted and shared.

## Code availability
Our code for running experiments and generating figures is publicly available at https://github.com/microsoft/vaccine_search_study.

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

## Acknowledgements

The authors thank Ruth Appel, Jonas Barklund, Lisa Cooper, Victor Dibia, Rahul Dodhia, Irena Gao, Kristina Gligorić, Kathleen Hall Jamieson, Ece Kamar, Paul Koch, Jure Leskovec, Besmira Nushi, Mayana Pereira, Emma Pierson, Yusuf Roohani, Rok Sosic, Albert Sun, Johan Ugander, Michihiro Yasunaga, and members of Johan Ugander's lab for helpful discussions and support. S.C. conducted this work as an intern at Microsoft.

## Author contributions

S.C. performed computational analysis. E.H. and S.C. conceived the top-level framing of the study. S.C., E.H., and A.F. jointly reviewed and engaged on methods, analyzed the results, and wrote the paper.

## Competing interests

The authors declare no competing interests.
