## [Peer Review File · Nature Communications]

Accurate measures of vaccination and concerns of vaccine holdouts from web search logsEditorial Note: This manuscript has been previously reviewed at another journal. This document only contains reviewer comments and rebuttal letters for versions considered at *Nature Communications*.

REVIEWERS' COMMENTS

Reviewer #1 (Remarks to the Author):

Thanks to the authors for their comprehensive revisions to the manuscript. I'm re-reviewing this paper, taking in to account my prior concerns, the authors' responses, and my understanding of the revised paper and its methods. I thought the methodology was robust from the start. The addition of the Granger causality and new methods strengthen a strong submission even more.

The authors read my concern correctly - the stated novelty of the findings and its clear communication in the manuscript for a general scientific audience. I'm happy with reframing the findings to emphasize the novelty of the outcomes, and I appreciate the care with which the authors have specified their findings and the clear "takeaways" we now have from the paper. I'm interested in vaccine holdout "phenotypes" that give quantitative evidence to why people are vaccine-hesitant - and therefore, what we can do from health policy to help persuade people to receive vaccines. I'm also happy to see the analysis of ZCTA differences about reasons for vaccine holdouts become more salient in this draft.

In addition to strengthening the novelty outright, these changes also improve the manuscript's positioning with other prior work on search data analysis. This is important because of previous studies on search data as a tool for health prediction and decision-making and the limits of search data (GFT to the Lazar et al. critique being one of the most prominent examples). I'm happy that the authors took these concerns to heart and have clarified places where their robust methods yielded exciting results. I'm excited that these results are valuable to the broader scientific community interested in search engine studies. (I'm less convinced by the methods arguments in 3 "New Methods for search log analysis" - people who can do analysis are internal to companies like Google and Microsoft, limiting the methods impact to folks with access to those datasets. Not that the methods gains aren't relevant for people in general, but that a narrow subset of people benefit from them.)

However, the reviewers have answered my questions in their response documents, well done! One request that I'd like to raise (and this is dependent on the editorial staff's discretion) is to incorporate these insights more heavily in the Introduction of the paper. The paper makes claims about differences between holdouts and profiles, and I would like more clarity about those "findings" are. This will help the authors communicate those results in the Introduction more quickly and successfully to readers and make their takeaways more salient.

The authors answered all my methods questions- which weren't many that needed changes, just improving the clarity of the writing and considering potential speedbumps in their analysis re: AMT, search data, how it's used, and its analysis.

Reviewer #2 (Remarks to the Author):

Thank you for addressing my comments and performing extra analyses. I think they have made the manuscript much stronger. I'd like to recommend accepting this article.

Reviewer #3 (Remarks to the Author):

The authors have submitted a revised version of the paper titled "Accurate Measures of Vaccination and Concerns of Vaccine Holdouts from Web Search Logs." This revised manuscript demonstrates significant improvements over the original, incorporating more rigorous statistical analyses, refined interpretations of the results, and stronger connections to the existing literature on the diffusion of innovation.

My primary concerns with the initial submission have been adequately addressed. Notably, the addition of a discussion that maps the taxonomy of search intents and users to the five-step decision-making process and the five adopter groups, as defined in Rogers' seminal work on diffusion of innovation, is particularly insightful. Furthermore, the inclusion of two Granger causality analyses and an inter-rater reliability test has significantly enhanced the robustness of the conclusions drawn. The authors' efforts to amend these aspects of the original manuscript are commendable.

I have two minor suggestions for further improvement:

- 1) The explanation provided in the authors' response concerning the choice between using a query-click graph per state and a large (weighted) national graph is insightful. It would be beneficial for this rationale to be included in the supporting information for readers.
- 2) In Table 5, which presents the Granger Causality analysis results, identifying the states that do not show significant results and exploring potential reasons for these anomalies could offer valuable insights.

These adjustments would enhance the clarity and impact of the research findings.

We thank the reviewers for re-reviewing our paper, and we were very glad to receive such positive reviews! We were happy to see that all reviewers felt like their concerns had been adequately addressed. We respond to the few remaining comments below:

Reviewer #1

Most of the reviewer's comments were positive and do not require a response. The review is quoted in its entirety here:

"Thanks to the authors for their comprehensive revisions to the manuscript. I'm re-reviewing this paper, taking in to account my prior concerns, the authors' responses, and my understanding of the revised paper and its methods. I thought the methodology was robust from the start. The addition of the Granger causality and new methods strengthen a strong submission even more.

The authors read my concern correctly - the stated novelty of the findings and its clear communication in the manuscript for a general scientific audience. I'm happy with reframing the findings to emphasize the novelty of the outcomes, and I appreciate the care with which the authors have specified their findings and the clear "takeaways" we now have from the paper. I'm interested in vaccine holdout "phenotypes" that give quantitative evidence to why people are vaccine-hesitant – and therefore, what we can do from health policy to help persuade people to receive vaccines. I'm also happy to see the analysis of ZCTA differences about reasons for vaccine holdouts become more salient in this draft.

In addition to strengthening the novelty outright, these changes also improve the manuscript's positioning with other prior work on search data analysis. This is important because of previous studies on search data as a tool for health prediction and decision-making and the limits of search data (GFT to the Lazar et al. critique being one of the most prominent examples). I'm happy that the authors took these concerns to heart and have clarified places where their robust methods yielded exciting results. I'm excited that these results are valuable to the broader scientific community interested in search engine studies. (I'm less convinced by the methods arguments in 3 "New Methods for search log analysis" – people who can do analysis are internal to companies like Google and Microsoft, limiting the methods impact to folks with access to those datasets. Not that the methods gains aren't relevant for people in general, but that a narrow subset of people benefit from them.)

*However, the reviewers have answered my questions in their response documents, well done! **One request that I'd like to raise** (and this is dependent on the editorial staff's discretion) is to incorporate these insights more heavily in the Introduction of the paper. The paper makes claims about differences between holdouts and profiles, and I would like more clarity about those "findings" are. This will help the authors communicate those results in the Introduction more quickly and successfully to readers and make their takeaways more salient.*

The authors answered all my methods questions- which weren't many that needed changes, just improving the clarity of the writing and considering potential speedbumps in their analysis re: AMT, search data, how it's used, and its analysis."

In response to the one request, we have edited the Introduction of the paper to more heavily emphasize differences between holdouts and to discuss the significance of those differences.

Reviewer #2

The reviewer's comments are entirely positive, quoted below:

"Thank you for addressing my comments and performing extra analyses. I think they have made the manuscript much stronger. I'd like to recommend accepting this article."

Reviewer #3

Most of the reviewer's comments were positive and do not require a response. The review is quoted in its entirety here:

"The authors have submitted a revised version of the paper titled "Accurate Measures of Vaccination and Concerns of Vaccine Holdouts from Web Search Logs." This revised manuscript demonstrates significant improvements over the original, incorporating more rigorous statistical analyses, refined interpretations of the results, and stronger connections to the existing literature on the diffusion of innovation.

My primary concerns with the initial submission have been adequately addressed. Notably, the addition of a discussion that maps the taxonomy of search intents and users to the five-step decision-making process and the five adopter groups, as defined in Rogers' seminal work on diffusion of innovation, is particularly insightful. Furthermore, the inclusion of two Granger causality analyses and an inter-rater reliability test has significantly enhanced the robustness of the conclusions drawn. The authors' efforts to amend these aspects of the original manuscript are commendable.

I have two minor suggestions for further improvement:

1) The explanation provided in the authors' response concerning the choice between using a query-click graph per state and a large (weighted) national graph is insightful. It would be beneficial for this rationale to be included in the supporting information for readers.

2) In Table 5, which presents the Granger Causality analysis results, identifying the states that do not show significant results and exploring potential reasons for these anomalies could offer valuable insights.

These adjustments would enhance the clarity and impact of the research findings."

In response to the two suggestions:

- 1) We have incorporated the explanation about why we used a query-click graph per state into the SI.
- 2) This is an interesting idea! Since we perform $50 \times 4 \times 3$ tests (per state, lag, and statistical test), we won't be able to list out all of the tests that do not show significant results, but we do definitely see that the comparisons of timeseries vary across states, both from the Granger causality analysis and the prior correlations. For example, in Figure S7 (the original Figure 11), we see that the correlations vary, eg, it is lower in Georgia, where the CDC line looks unusually bumpy, and much lower in North Carolina, where the CDC line exhibits a very anomalous spike in July. We also see in Figure S7 that the optimal lag, ie, the one that achieves the maximum correlation, varies between states, which could also explain why, in the Granger causality test, one lag might be significant for one state but not another.